# ICAM-1 nanoclusters regulate hepatic epithelial cell polarity by leukocyte adhesion-independent control of apical actomyosin

Cristina Cacho-Navas[1], Carmen López-Pujante[1], Natalia Reglero-Real[2], Natalia Colás-Algora[1], Ana Cuervo[3], Jose Javier Conesa[3], Susana Barroso[1], Gema de Rivas[1], Sergio Ciordia[3], Alberto Paradela[3], Gianluca D'Agostino[3], Carlo Manzo[4], Jorge Feito[5], Germán Andrés[1], Francisca Molina-Jiménez[6,7], Pedro Majano[6,8,9], Isabel Correas[1], José-Maria Carazo[3], Sussan Nourshargh[2], Meritxell Huch[10], Jaime Millán[1]*

[1]Centro de Biologia Molecular Severo Ochoa, CSIC-UAM, Madrid, Spain; [2]William Harvey Research Institute, Faculty of Medicine and Dentistry, Queen Mary University of London, London, United Kingdom; [3]Centro Nacional de Biotecnologia (CSIC), Madrid, Spain; [4]Facultat de Ciències, Tecnologia i Enginyeries, Universitat de Vic – Universitat Central de Catalunya (UVic-UCC), Vic, Spain; [5]Servicio de Anatomía Patológica, Hospital Universitario de Salamanca, Salamanca, Spain; [6]Molecular Biology Unit, Hospital Universitario de la Princesa, Madrid, Spain; [7]Instituto de Investigación Sanitaria Hospital Universitario de La Princesa (IIS-Princesa), Madrid, Spain; [8]Centro de Investigación Biomédica en Red de Enfermedades Hepáticas y Digestivas (CIBERehd), Madrid, Spain; [9]Department of Cellular Biology, Universidad Complutense de Madrid, Madrid, Spain; [10]Max Planck Institute of Molecular Cell Biology and Genetics, Dresden, Germany

*For correspondence:
jmillan@cbm.csic.es

Competing interest: The authors declare that no competing interests exist.

**Abstract** Epithelial intercellular adhesion molecule (ICAM)-1 is apically polarized, interacts with, and guides leukocytes across epithelial barriers. Polarized hepatic epithelia organize their apical membrane domain into bile canaliculi and ducts, which are not accessible to circulating immune cells but that nevertheless confine most of ICAM-1. Here, by analyzing ICAM-1_KO human hepatic cells, liver organoids from ICAM-1_KO mice and rescue-of-function experiments, we show that ICAM-1 regulates epithelial apicobasal polarity in a leukocyte adhesion-independent manner. ICAM-1 signals to an actomyosin network at the base of canalicular microvilli, thereby controlling the dynamics and size of bile canalicular-like structures. We identified the scaffolding protein EBP50/NHERF1/SLC9A3R1, which connects membrane proteins with the underlying actin cytoskeleton, in the proximity interactome of ICAM-1. EBP50 and ICAM-1 form nano-scale domains that overlap in microvilli, from which ICAM-1 regulates EBP50 nano-organization. Indeed, EBP50 expression is required for ICAM-1-mediated control of BC morphogenesis and actomyosin. Our findings indicate that ICAM-1 regulates the dynamics of epithelial apical membrane domains beyond its role as a heterotypic cell–cell adhesion molecule and reveal potential therapeutic strategies for preserving epithelial architecture during inflammatory stress.

## eLife assessment

The authors report **useful** findings on the novel function of apical ICAM-1 in regulating bile duct homeostasis in the liver. The strength of evidence is **solid** using appropriate methodology with only minor weakness. The findings will be of interest to researchers in hepatology and membrane traffic biology.

## Introduction

Intercellular adhesion molecule (ICAM)-1 is the counterreceptor of leukocyte β2-integrins and mediates firm adhesion of leukocytes to epithelial and endothelial cells (*Rothlein et al., 1986*; *Sumagin et al., 2014*; *Reglero-Real et al., 2012*; *Vestweber, 2015*). ICAM-1 is not only a passive endothelial surface anchor for circulating immune cells, but also signals and remodels the plasma membrane and the underlying actin cytoskeleton to promote transendothelial migration (TEM) and extravasation of leukocytes in many inflammatory diseases (*Reglero-Real et al., 2012*), including those involving cholestasis, in which inhibition of ICAM-1 function reduces liver injury (*Gujral et al., 2004*). ICAM-1 is apically confined in polarized intestinal and hepatic epithelia (*Sumagin et al., 2014*; *Reglero-Real et al., 2014*; *Reglero-Real et al., 2016*), which confers on these cellular barriers the capacity to establish a haptotactic gradient between apical and basolateral membrane domains in order to guide infiltrated immune cells (*Sumagin et al., 2014*). The apical membrane domains of hepatocytes and cholangiocytes form bile canaliculi and bile ducts, respectively, which drain bile acids and other hepatic molecules into the gastrointestinal tract. The composition of the apical plasma membrane domains in these small channels differs from that of basolateral membranes and from plasma membranes in non-polarized cells, and is determined by an actin cytoskeletal network, which concentrates in these apical domains, and by mechanisms of polarized intracellular sorting of proteins and lipids toward these membrane regions (*Meyer et al., 2020*; *Tsukada et al., 1995*; *Fu et al., 2010*; *Cacho-Navas et al., 2022*; *Müsch, 2014*). The filamentous actin cytoskeleton in the bile canaliculus forms small membrane protrusions, called microvilli, which contain actin but not myosin. However, early electron microscopy analyses and studies of bile canaliculus dynamics in vivo revealed the additional existence of an acto-myosin mesh surrounding the basal regions of these microvilli. This cytoskeletal scaffold regulates the diameter and dynamics of the biliary network and is involved in bile trafficking and cholestasis (*Meyer et al., 2020*; *Tsukada et al., 1995*; *Gupta et al., 2017*).

The population of hepatic ICAM-1 molecules that are segregated into bile canaliculi and ducts is not accessible to circulating immune cells, which are in contact with the basolateral membranes of hepatic epithelial cells through the space of Disse. Indeed, the adhesion of leukocytes increases when these epithelial cells lose their polarity and expose this receptor pool. Studies of ICAM-1 dynamics have demonstrated that basolateral ICAM-1 is highly dynamic, but the apical pool of receptors is highly stable and confined within BCs (*Reglero-Real et al., 2014*). This confinement depends on the interaction of ICAM-1 with the underlying canalicular F-actin (*Reglero-Real et al., 2014*). However, it is currently unknown whether segregation and immobilization of ICAM-1 into these hepatic apical plasma membrane domains play any role in canalicular dynamics and function.

Here, we have followed experimental strategies that facilitate the analysis of ICAM-1 function in the absence of interactions with circulating immune cells. We have generated and characterized CRISPR-CAS9-edited human hepatic epithelial cells lacking ICAM-1 expression, and liver organoids derived in 3D from hepatic bipotent ductal stem cells of wildtype (WT) and ICAM-1_KO mice. We show that ICAM-1 regulates the size and dynamics of bile canalicular-like structures (BCs) in the absence of leukocyte adhesion by controlling the actomyosin meshwork that surrounds these apical lumens. Hepatic epithelial cells lacking ICAM-1 expression have their BCs massively enlarged, although these are properly sealed and retain their ability to confine other canalicular proteins. In contrast, upregulation of ICAM-1 expression by inflammatory cytokines greatly reduced BC frequency in a myosin-II-dependent manner. By combining super-resolution microscopy and proximal interactomics analyses, we also reveal that microvillar ICAM-1 is organized into membrane nanoclusters with EBP50, a membrane-cytoskeleton protein connector that has been linked to canalicular disruption during intrahepatic cholestasis (*Bryant et al., 2014*; *Reczek and Bretscher, 2001*; *Hsu et al., 2010*; *Li et al., 2015*). Indeed, we show that EBP50 controls ICAM-1-mediated regulation of polarity and actomyosin. Our findings indicate that ICAM-1 acts not only as a mediator of leukocyte infiltration in inflammatory

and cholestatic diseases, but also as a master regulator of cellular scaffolds that maintain the apical membranes that form the biliary network.

## Results

### ICAM-1 regulates the size and morphology of apical canalicular-like structures in human hepatic epithelial cells

We had previously shown that ICAM-1 is expressed in sinusoids but also concentrates in the bile canaliculi and bile ducts of polarized hepatic epithelial cells of the human liver parenchyma (*Figure 1A*; *Reglero-Real et al., 2014*). HepG2 cells are spontaneously polarized human hepatic epithelial cells that form actin-rich, BCs and are thus a prototypical in vitro model for studying hepatic apicobasal polarity (*Reglero-Real et al., 2014*; *Lázaro-Diéguez and Müsch, 2017*; *Madrid et al., 2010*; *van IJzendoorn et al., 1997*; *Cacho-Navas et al., 2022*). To investigate the role of ICAM-1 on hepatic epithelial cell polarity, we first generated HepG2 cells in which the *ICAM1* gene was edited using CRISPR/CAS9 (ICAM-1_KO cells) (*Figure 1B and C*). ICAM-1_KO cells proliferated slightly slower than parental WT cells and had similar size and percentage of cell death (*Figure 1—figure supplement 1A and B*). In addition, ICAM-1_KO cells exhibited larger, less spherical and more elongated BCs than did parental WT cells, suggesting a role for this receptor in regulating BC morphology (*Figure 1C and D*). We called these structures enlarged BCs (eBCs) because their area was more than 2.5 times that of BCs in WT cells, although the frequency of eBCs per cell was similar to that of BCs (*Figure 1E*). Similar to BCs, eBCs were enriched in microvilli, sealed by tight junctions (TJs), and contained intracellular canalicular markers such as radixin (*Figure 1—figure supplement 1C*). In addition, eBC did not show significant alterations in the distribution of membrane proteins that reach BCs by a direct route of intracellular transport, such as MDR1 and MRP2 (*Figure 1D*, *Figure 1—figure supplement 1C*; *Sai et al., 1999*), or by an indirect route, such as CD59 (*Figure 1D*; *de Marco et al., 2002*). eBCs were also surrounded by a subapical compartment (SAC), as observed by detecting endogenous plasmolipin (*Cacho-Navas et al., 2022*; *Figure 1—figure supplement 1D*) or by expressing the SAC marker GFP-Rab11 (*Figure 1—figure supplement 1E*). The enlargement of these canalicular structures was rescued by expressing ICAM-1-GFP in ICAM-1_KO cells, which reduced the size of BCs to those in control HepG2 cells (*Figure 1F*). ICAM-1 knockdown (KD) also increased the BC area, confirming the effect of ICAM-1 reduction on BC size (*Figure 1—figure supplement 1F and G*). To analyze ICAM-1-mediated morphological changes further, we performed correlative cryo-soft X-ray tomography (cryo-SXT) in control and ICAM-1_KO HepG2 cells that stably expressed GFP-Rab11 to enable the localization of BCs in live cells. Correlative Cryo-SXT revealed that eBCs from ICAM-1_KO cells contained all the morphological features of BCs, although in an enlarged state, including the abundance and length of the microvilli (*Figure 1G*).

Potential fusion events between eBCs were also observed by Cryo-SXT (*Figure 2A*) and by time-lapse fluorescence microscopy (*Figure 2B*, left panel, central graph, *Videos 1 and 2*) in ICAM-1_KO cells. However, no differences in the relative changes in canalicular areas between frames were observed between KO and WT cells in these time-lapse experiments (*Figure 2B*, right graph). Indeed, long-term culture of ICAM-1_KO (*Figure 2C*, *Figure 2—figure supplement 1A*) and ICAM-1_KD (*Figure 2—figure supplement 1B*) cells induced the appearance of microvillus-rich structures that were much larger than eBCs observed between 48 and 72 hr, covering areas of up to 100 µm². These structures were surrounded by TJs with a cobblestone morphology, although the TJ polygons did not enclose an underlying nucleus and, hence, did not correspond to single cells establishing cell–cell contacts (*Figure 2C and D*). Spherical, canalicular-like domains enriched in F-actin (*Figure 2D*, *Figure 2—figure supplement 1C*) and similar to the BCs observed in control cells were found in contact with these large canalicular structures, suggesting that the fusion of several BCs might contribute to the enlargement of canalicular domains in ICAM-1-depleted cells. Finally, to test whether these massive eBCs were exposed to the extracellular milieu, WT and ICAM-1_KO cells were incubated in the cold with sulfo-NHS-biotin, which labels lysine residues from proteins that are exposed on the cell surface and in contact with the medium. Cells were fixed and biotinylated proteins stained with TRITC-conjugated neutravidin. This revealed that streptavidin did not penetrate the lumens of the eBCs, indicating that these large structures are not exposed to the extracellular

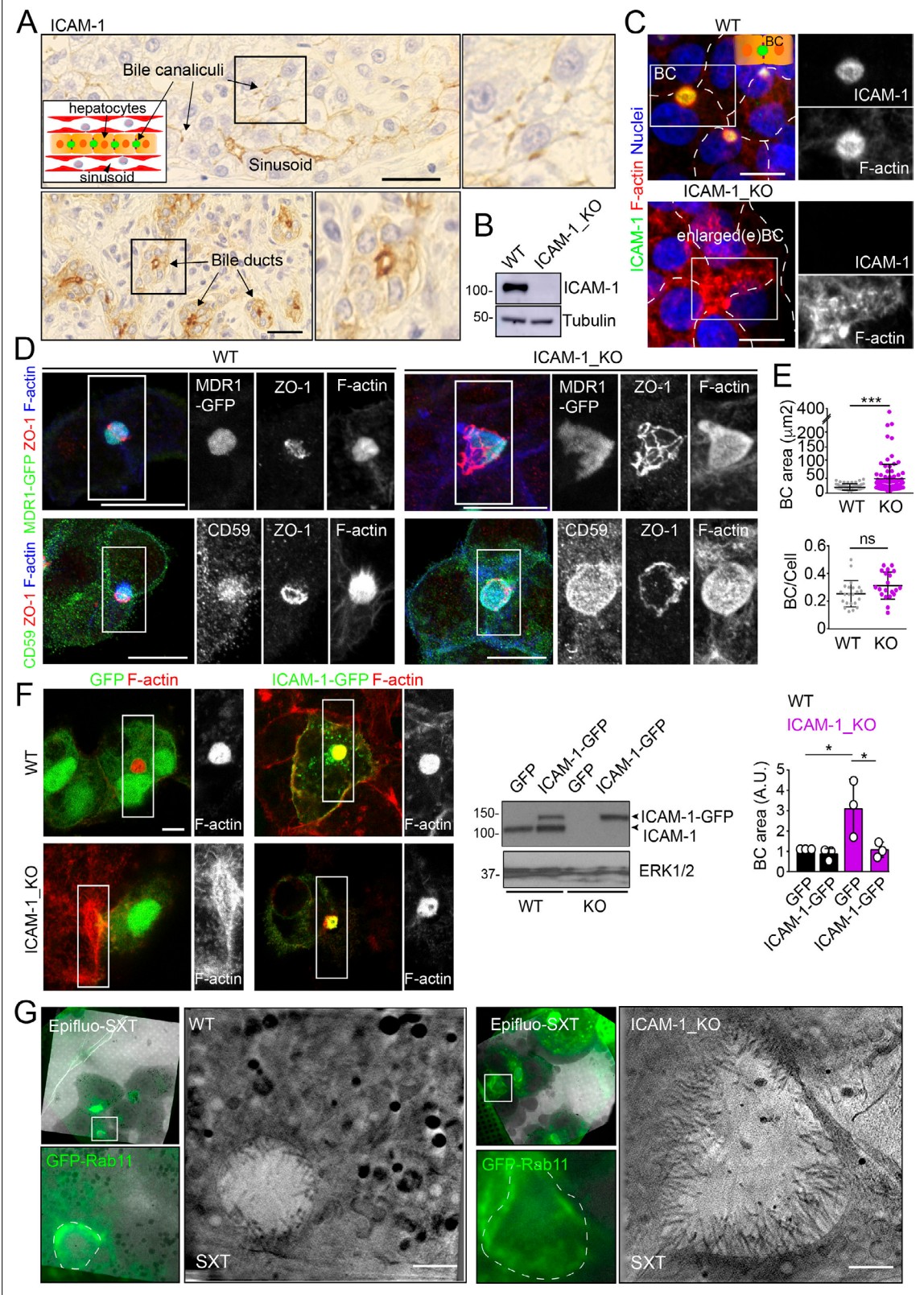

**Figure 1.** Intercellular adhesion molecule-1 (ICAM-1) regulates the size of apical bile canaliculi (BCs) in polarized HepG2 cells. (**A**) ICAM-1 concentrates in bile canaliculi (top images) and bile ducts (bottom images) from polarized hepatocytes and cholangiocytes, respectively (arrows), and in sinusoids in human livers from healthy donors. Right images show a twofold enlargement of the boxed areas in the left images. Inset cartoon represents the basic organization of polarized hepatocytes at the hepatic lobule. Their apical membranes form bile canaliculi and their basolateral membranes face the

*Figure 1 continued on next page*

Figure 1 continued

sinusoids. Scale bars, 10 μm (**B**) CRISPR-CAS9-mediated editing of the *ICAM1* gene (ICAM-1_KO) in HepG2 cells. Several clones were selected and pooled to prevent clonal variations. WT, parental wild-type cells. (**C, D**) Polarized human epithelial HepG2 cells form BCs. Control WT and ICAM-1_KO HepG2 cells were cultured on coverslips for 72 hr, fixed and stained for F-actin and ICAM-1, which concentrate in apical BCs (**C**), for ZO-1 and F-actin in cells expressing MDR1-GFP (**D**, top images) and for CD59, ZO1 and F-actin (**D**, bottom images). Scale bars, 20 μm. (**E**) Plots represent the mean ± SD. BC area increased from 18.9 ± 1.1 μm² in WT cells to 51.2 ± 7.2 μm² in ICAM-1_KO cells. ***p<0.001; ns, not statistically significant. BC area and frequency were quantified in at least 70 cells per experiment. Nuclei were stained with DAPI. (**F**) WT and ICAM-1_KO cells were transfected with GFP or ICAM-1-GFP expression plasmids, grown for 48 hr, and processed for immunofluorescence and confocal microscopy analysis (left images) or lysed for western blot analysis (central images). Single channels from the boxed areas are enlarged on the right of the corresponding image. The right panel shows the quantification of BC areas and represents the mean ± SD. * p<0.05. n = 3. Scale bar, 10 μm. (**G**) Cryo-SXT of WT and ICAM-1_KO cells stably expressing GFP-Rab11 to localize BCs. Cells were cultured on holey carbon grids for TEM for 48–72 hr. BCs were identified in live cells with an epifluorescence microscope (dotted lines) and immediately vitrified and cryopreserved for cryo-soft X-ray tomography (cryo-SXT). SXT images show a slice of the reconstructed tomogram from the boxed areas in the correlative Epifluo-SXT images. GFP-Rab11 panels display the epifluorescence images of the same area. Scale bars, 2 μm.

The online version of this article includes the following source data and figure supplement(s) for figure 1:

**Source data 1.** Original file for the western blot analysis in *Figure 1B*.

**Source data 2.** PDF containing *Figure 1B* and original scans of the relevant western blot analysis (anti-ICAM-1 and anti-tubulin), with the highlighted bands squared.

**Source data 3.** Original file for the western blot analysis in *Figure 1F*.

**Source data 4.** PDF containing *Figure 1F* and original scans of the relevant western blot analysis (anti-ICAM-1 and anti-ERK), with the highlighted bands squared.

**Figure supplement 1.** Effect of ICAM-1 depletion on cell apicobasal polarity, proliferation, size and survival.

**Figure supplement 1—source data 1.** Original file for the western blot analysis in *Figure 1—figure supplement 1F* (anti-ICAM-1 and anti-ERK).

**Figure supplement 1—source data 2.** PDF containing *Figure 1—figure supplement 1F* and original scans of the relevant western blot analysis (anti-ICAM-1 and anti-ERK), with the highlighted bands squared.

milieu (*Figure 2E*). Taken together, our analyses indicate that eBCs fuse more frequently than BCs and become considerably expanded after 4 d of polarization.

## ICAM-1 upregulation by inflammatory cytokines reduces hepatic apicobasal polarity

ICAM-1 editing or silencing increases the size of apical surfaces while maintaining the frequency of BCs, indicating that reducing this receptor promotes apicobasal polarity in hepatic epithelial cells. ICAM-1 is basally expressed in hepatic epithelial cells in vitro and in vivo (*Smith and Thomas, 1990*; *Adams et al., 1989*; *Park et al., 2010*), but its expression is strongly upregulated by cytokines during the inflammatory response (*Volpes et al., 1990*; *Kvale and Brandtzaeg, 1993*). We next addressed the effect of exposing polarized WT and ICAM-1_KO cells to a set of cytokines involved in liver inflammation (*Park et al., 2010*; *Lacour et al., 2005*; *He et al., 2021*). IL-1β, IFN-γ, and the combination of TNF-α and IFN-γ induced the expression of ICAM-1 protein by a factor of 4–6, whereas TNF-α had a moderate effect (*Figure 3A*). The quantification of the apical-to-basolateral ratio revealed a greater increase in expression at the basolateral domains compared to the apical ones (*Figure 3B and C*). These changes in expression and distribution were correlated with a reduction in the number of BCs in cytokine-stimulated cells (*Figure 3D*). However, BC loss was prevented when ICAM-1_KO cells were stimulated with these cytokines (*Figure 3D*, left graphs). In addition, BC size was not reduced upon cytokine stimulation (*Figure 3D*, right graphs), indicating that either the morphological remodeling of the remaining BCs is controlled by additional expression changes and signaling events triggered by these cytokines or that these inflammatory mediators provoke the collapse of the smallest BCs, so only the enlarged ones are preserved after the treatment, giving rise to a moderate increase in the average size. However, collectively, these findings demonstrate that ICAM-1 expression levels modulate BC size and/or frequency and thus have a role in hepatic epithelial apicobasal polarity.

## ICAM-1 controls canalicular actomyosin, thereby regulating BC size

Upon leukocyte adhesion, ICAM-1 signals to actomyosin in endothelial cells (*Reglero-Real et al., 2012*; *Millán and Ridley, 2005*; *van Buul and Hordijk, 2004*; *van Buul et al., 2007*). We hypothesized that

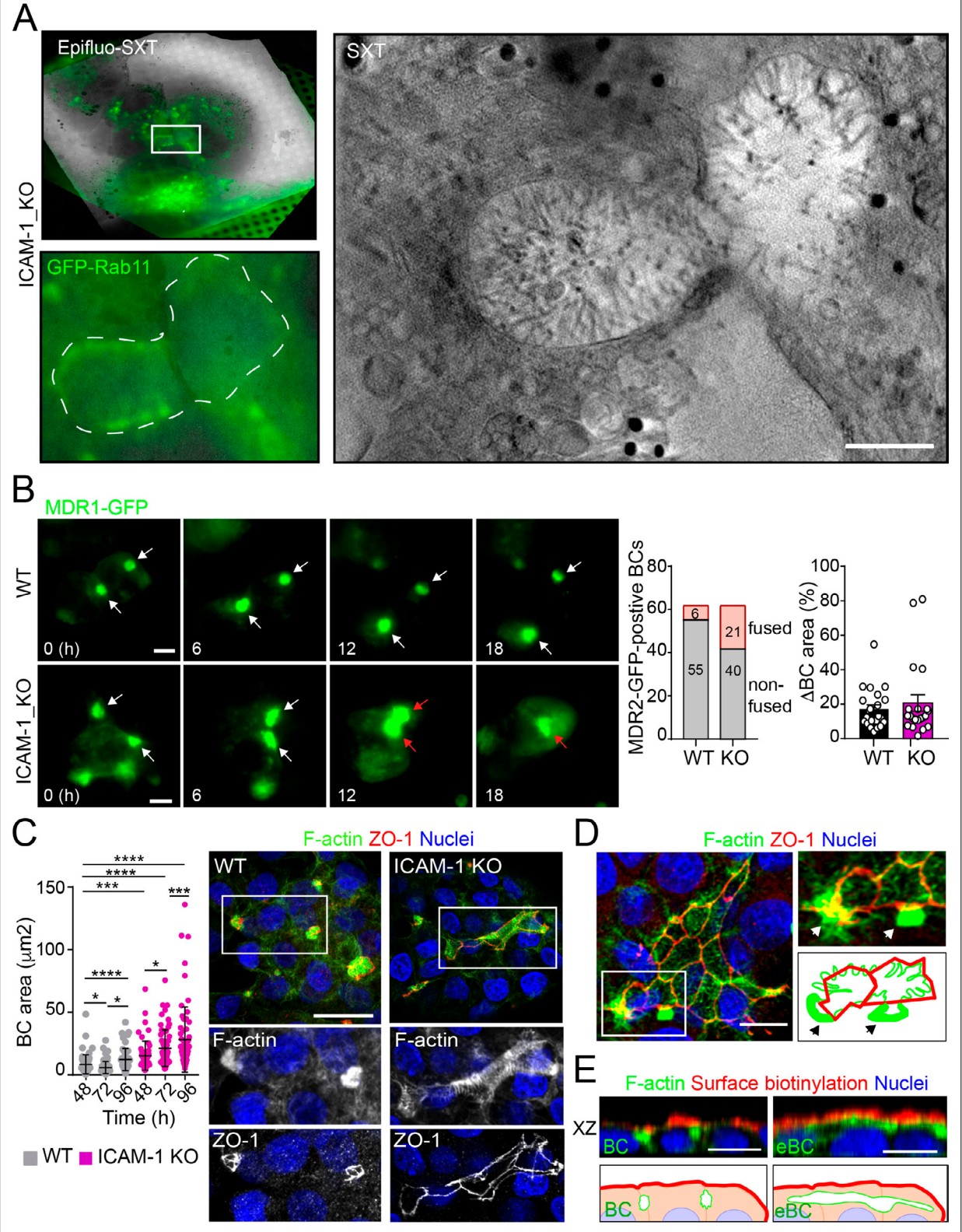

**Figure 2.** Intercellular adhesion molecule-1 (ICAM-1) regulates bile canaliculi (BC) dynamics in polarized HepG2 cells. (**A**) Correlative cryo-soft X-ray tomography (cryo-SXT) of ICAM-1_KO cells stably expressing GFP-Rab11 showing two fusing BCs. The SXT image shows a slice of the reconstructed tomogram from the boxed area in the correlative Epifluo-SXT image (top left). The GFP-Rab11 panel displays the epifluorescence image of the same area. Scale bar, 2 μm. (**B**) HepG2 cell stably expressing MDR1-GFP were subjected to time-lapse fluorescence microscopy analysis. White arrows point

*Figure 2 continued on next page*

*Figure 2 continued*

at MDR1-GFP-positive BCs. Red arrows point at two fusing MDR-1-GFP-positive BCs. Central panel shows the quantification of BCs in which fusing events were observed during the time-lapse microscopy assays. Right panel quantifies variations in BC area between consecutive frames, expressed as positive values. Scale bars, 10 μm (**C**) WT and ICAM-1_KO hepatic cells were cultured for at least 96 hr, fixed and stained for the indicated proteins. Scale bar, 20 μm. The left scatterplot shows the quantification of BC area of cells cultured for the indicated periods and represents the mean ± SD. *p<0.05, ***p<0.001, ****p<0.0001. 50 BCs per experiment, n = 3. (**D**) Distribution of F-actin and ZO-1 in ICAM-1_KO cells cultured for 96 hr suggests coalescence of small BCs into enlarged (e)BC. Enlarged areas show F-actin-enriched BCs (arrows) in contact with massive eBCs. Bottom-right images show a graphical representation of the top-right images. Scale bar, 10 μm. (**E**) ICAM-1_KO cells were grown for 96 hr and incubated with sulfo-NHS-biotin for 30 min at 4°C, washed, fixed, and permeabilized. Biotinylated surface proteins were detected with TRITC-conjugated streptavidin. XZ stack projections are shown to visualize the relative localization of surface proteins, accessible to sulfo-NHS-biotin from the extracellular milieu, and F-actin-rich BCs (left images) and eBC (right images), which are sealed and not accessible from the extracellular milieu. Nuclei were stained with DAPI. Bottom images show a graphical representation of the top images. Scale bars, 10 μm.

The online version of this article includes the following source data and figure supplement(s) for figure 2:

**Figure supplement 1.** Long-term ICAM-1 depletion induces massively enlarged BCs.

**Figure supplement 1—source data 1.** Original file for the western blot analysis in *Figure 2—figure supplement 1B* (anti-ICAM-1 and anti-tubulin).

**Figure supplement 1—source data 2.** PDF containing *Figure 2—figure supplement 1B* and original scans of the relevant western blot analysis (anti-ICAM-1 and anti-tubulin), with the highlighted bands squared.

hepatic ICAM-1, which is abundant and highly confined within BCs (*Reglero-Real et al., 2014*), could signal locally to actomyosin and thereby regulate the size and dynamics of these apical membrane domains in the absence of any interaction with immune cells. To analyze the distribution of actomyosin, polarized hepatic epithelial cells expressing GFP-tagged myosin light chain (GFP-MLC) were incubated with SirActin, which concentrated into BCs similar to the phalloidin staining (*Figures 4A and 1C*). GFP-MLC formed a spherical network that surrounded SirActin-positive BCs (*Figure 4A*, *Video 3*). Single planes of the confocal time-lapse projections showed that MLC formed a ring at the base of microvilli-like actin domains, but did not localize in the most central parts of microvillar structures. High-resolution confocal microscopy of endogenous F-actin and myosin light chain phosphorylated in the T18/S19 residues (p-MLC) confirmed that active non-muscle myosin-II complexes were distributed in the most distal parts BC where they overlapped with F-actin, which also formed microvilli-like structures (*Figure 4B*). In addition, the regions of F-actin that overlapped with ICAM-1 and the actomyosin network clearly differed, and the co-localization of p-MLC and ICAM-1 staining was much less extensive than that of these two

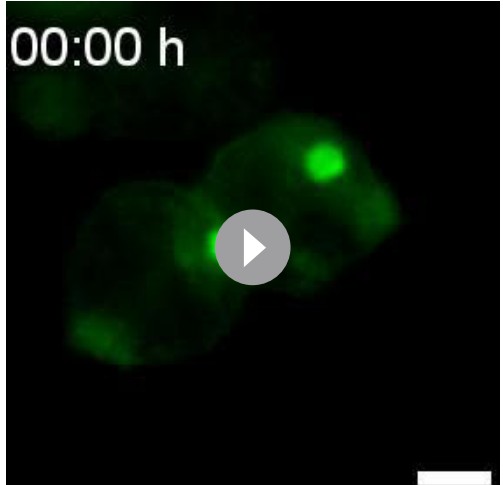

**Video 1.** Time-lapse fluorescence microscopy of polarized WT HepG2 cells stably expressing MDR1-GFP. Images were acquired at 15 min intervals for 18 hr and displayed at four frames per second. Note the fusion of two bile canaliculi (BC) between 14 and 18 hr. Scale bar, 10 μm.

https://elifesciences.org/articles/89261/figures#video1

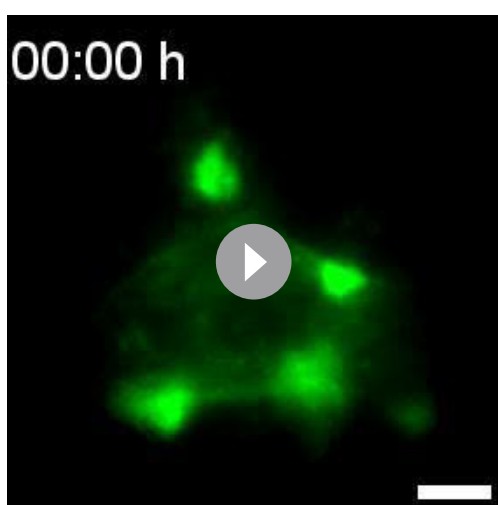

**Video 2.** Time-lapse fluorescence microscopy of polarized ICAM-1_KO HepG2 cells stably expressing MDR1-GFP. Images were acquired at 15 min intervals for 18 hr and displayed at four frames per second. Scale bar, 10 μm.

https://elifesciences.org/articles/89261/figures#video2

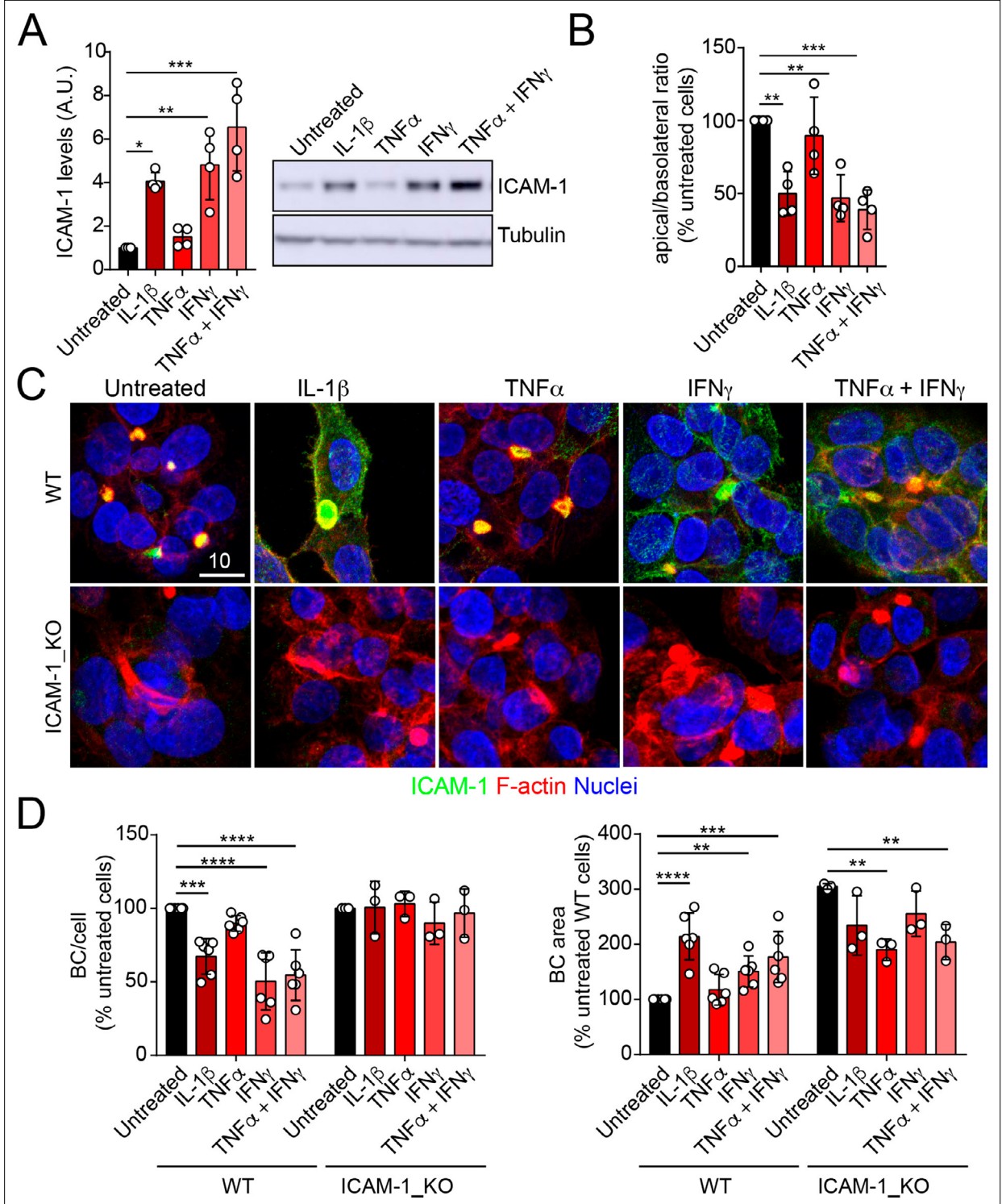

**Figure 3.** Upregulation of intercellular adhesion molecule-1 (ICAM-1) expression in response to inflammatory cytokines reduces bile canaliculi (BC) frequency. (**A–D**) Polarized hepatic epithelial cells were stimulated or not for 48 hr with 50 ng/ml TNF-α, 15 ng/ml IL-1β, and IFN-γ 1000 U/ml as indicated. (**A**) Cells were lysed and ICAM-1 expression levels detected by western blot. Tubulin was blotted as a loading control. (**B–D**) Cells were fixed and stained for ICAM-1, F-actin, and nuclei. (**B**) Basolateral-to-apical ratio of ICAM-1 staining intensities. (**C**) Representative confocal images of the effect of cytokine treatment. Scale bar, 10 μm. (**D**) Quantification of BCs (left) and BC area (right). Bars represent the mean ± SD. n ≥ 3. *p<0.05, **p<0.01, ***p<0.001, ****p<0.0001.

The online version of this article includes the following source data for figure 3:

*Figure 3 continued on next page*

*Figure 3 continued*

**Source data 1.** Original file for the western blot analysis in *Figure 3A*.

**Source data 2.** PDF containing *Figure 3A* and original scans of the relevant western blot analysis (anti-ICAM-1 and anti-tubulin), with the highlighted bands squared.

proteins with F-actin, as shown by Manders' analyses of the confocal images (*Figure 4B*, central and right panels). Importantly, the canalicular stainings of p-MLC (*Figure 4C*) and of myosin heavy chain-IIb (MHCIIb) (*Figure 4D*) were significantly reduced in ICAM-1_KO cells, whereas the F-actin levels were maintained (*Figure 4E*). In contrast, F-actin and p-MLC distribution in the basolateral membrane domains were higher in ICAM-1_KO cells than that in WT cells (*Figure 4E*, *Figure 4—figure supplement 1A*). The exogenous expression of ICAM-GFP in ICAM-1_KO cells restored the canalicular actomyosin network, indicating that ICAM-1 regulates the actomyosin cytoskeleton surrounding BCs (*Figure 4C–E*, *Figure 4—figure supplement 1A*). Moreover, cells treated with blebbistatin, the ATPase inhibitor for non-muscle myosin-II, induced eBCs in a manner similar to that observed in ICAM-1_KO cells (*Figure 4F*). Blebbistatin also prevented the ICAM-1-dependent loss of BCs in response to IL-1β (*Figure 4G*).

MLC phosphorylation and actomyosin contraction are regulated by the Rho kinases (ROCKs), major effectors of the RhoA subfamily of GTPases. ROCK inhibition with Y27632 phenocopied the effect of ICAM-1 editing and blebbistatin treatment on BC size (*Figure 4—figure supplement 1B*). The RhoA subfamily comprises the small GTPases RhoA, RhoB, and RhoC (*Hodge and Ridley, 2016*; *Riento and Ridley, 2003*), which are activated and translocated to the plasma membrane in response to different stimuli (*Marcos-Ramiro et al., 2016*; *Marcos-Ramiro et al., 2014*). The expression of GFP-RhoB and GFP-RhoC in polarized hepatic cells suggested that the RhoA subfamily significantly concentrates in the plasma membrane of BCs (*Figure 4—figure supplement 1C*). GFP-RhoA expression completely depolarized these hepatic epithelial cells, so its apicobasal distribution could not be addressed (not shown), whereas another Rho GTPase, GFP-Rac1, was almost evenly distributed between canalicular and basolateral plasma membrane domains (*Figure 4—figure supplement 1C*). Together, these results indicate that BCs concentrate the machinery regulating the Rho-ROCK-actomyosin signaling pathway.

## ICAM-1 signaling to actin and non-muscle myosin-II prevents BC morphogenesis

To address whether hepatic ICAM-1 signaling to actomyosin affects hepatic cell polarity, while taking into account that canalicular ICAM-1 cannot be accessed from the canalicular lumen (*Figure 2E*), we adopted an alternative strategy, which involved investigating the effect of ICAM-1 surface immobilization on BC morphogenesis of epithelial cells that initially were non-polarized. A variety of antibodies, including anti-ICAM-1 antibody, and fibronectin were immobilized on coverslips. Cells were then trypsinized and seeded on the antibody-coated coverslips to analyze the effect of ICAM-1-mediated signaling. Cells seeded on anti-ICAM-1 antibody-coated coverslips were unable to form BCs after 24 hr of culture and had more stress fibers than did cells seeded on surfaces coated with control immunoglobulin(Ig)-G, anti-transferrin receptor (TfR) or the integrin ligand, fibronectin, all of which formed BCs with a frequency similar to that of cells plated on uncoated surfaces (*Figure 5A–C*). This result indicated that receptor engagement signaled to F-actin, thereby preventing the acquisition of polarity. Moreover, cells on anti-ICAM-1 antibodies frequently displayed stellate stress fibers that were similar to those arising in response to the expression of the constitutively active mutant of ROCK (*Figure 5A–C*; *Garg et al., 2008*; *Nakano et al., 1999*). We found that in addition to the fraction of receptor localized on the basal surface after 24 hr of adhesion, stellate stress fibers concentrated surface ICAM-1 in their proximities (*Figure 5A*, *Figure 5—figure supplement 1A*), whereas p-MLC localized at the F-actin-rich regions where stellate stress fibers arise (*Figure 5B*, *Figure 5—figure supplement 1A*). These cellular structures did not constitute aberrant BCs since they neither had a canalicular morphology nor concentrated other canalicular markers such as MDR1 (*Figure 5D*). When cells seeded on anti-ICAM-1-coated coverslips were treated with blebbistatin, canalicular morphogenesis was restored to a level comparable with that observed in ICAM-1_KO cells on anti-ICAM-1 antibody-coated coverslips or in cells cultured on control coverslips (*Figure 5B and C*, *Figure 5—figure*

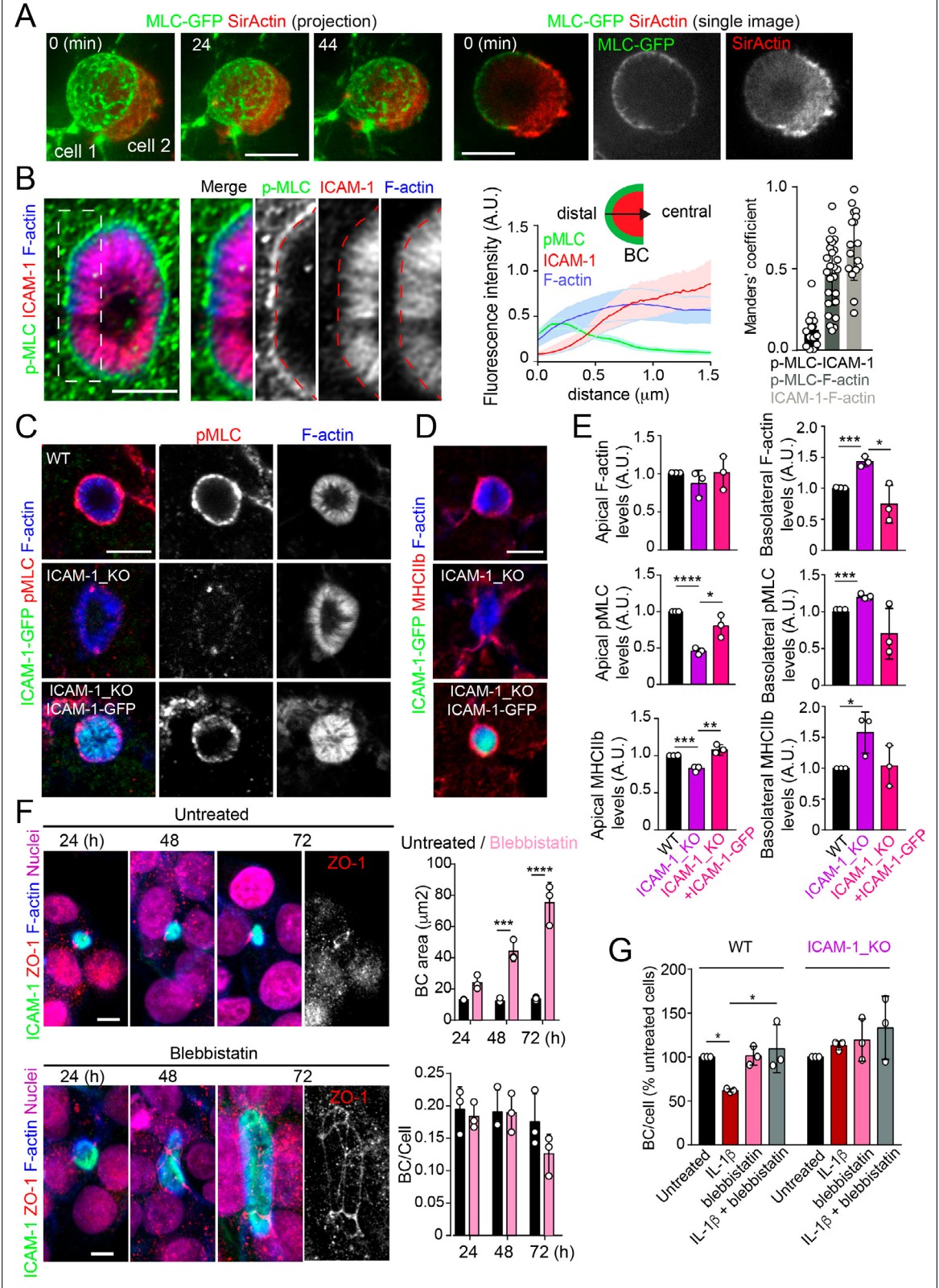

**Figure 4.** Intercellular adhesion molecule-1 (ICAM-1) controls canalicular membrane expansion by regulating a distal canalicular actomyosin network. (**A, B**) Bile canaliculi (BC) contain a ring of actomyosin. (**A**) Polarized HepG2 cells expressing GFP-MLC were incubated with SirActin for 2 hr and subjected to time-lapse confocal microscopy. Left images show the image projection of the indicated frames. Right images show a single image of the stack corresponding to t = 0 of the same time-lapse video. (**B**) Polarized hepatic epithelial cells were cultured on coverslips for 72 hr, fixed and

*Figure 4 continued on next page*

*Figure 4 continued*

stained for F-actin, ICAM-1, and phosphorylated myosin light chain (pMLC) (left images). Central panel: relative distribution of the staining intensity for the indicated proteins, starting from the distal parts of BCs, which contain actomyosin, toward the centers of the BCs, which contain the microvillar tips. Note that p-MLC staining is mostly distal, F-actin distributed all along the profile, and ICAM-1 is mostly found in microvilli. Right panel: Manders' analysis for the indicated pairs of staining. Scale bar, 3 µm. (**C, D**) Control WT, ICAM-1_KO, and ICAM-1_KO cells expressing ICAM-1-GFP were cultured on coverslips for 72 hr, fixed and stained for pMLC and F-actin (**C**) and for F-actin and non-muscle myosin heavy chain-IIb (MHCIIb) (**D**). Scale bar, 5 µm. (**E**) Quantification of relative apical and basolateral levels of F-actin, pMLC, and MHCIIb detected in (**C**) and (**D**). n = 3. At least 20 BCs (apical) or 20 cells (basolateral) were quantified in each experiment. (**F**) WT cells were exposed or not to 10 µM of the myosin inhibitor blebbistatin for the indicated times. Scale bar, 10 µm. Plots show the quantification of the effect of blebbistatin on BC size (top) and BC frequency (bottom). n = 3. At least 20 BCs were quantified in each experiment. (**G**) WT and ICAM-1_KO cells were cultured, stimulated with IL-1β, fixed and stained as in *Figure 3*. When indicated, cells were incubated with 10 µM blebbistatin for the last 40 hr. BC frequency was quantified. Quantifications represent the mean ± SD. n ≥ 3. *p<0.05, **p<0.01, ***p<0.001, ****p<0.0001. Except otherwise indicated, scale bars correspond to 5 µm.

The online version of this article includes the following figure supplement(s) for figure 4:

**Figure supplement 1.** ROCK inhibition phenocopies the effect of ICAM-1 depletion and myosin-II inhibition on BC size.

*supplement 1B*). This implies that the impairment of BC formation in response to surface ICAM-1 immobilization depends on non-muscle myosin-II. As previously described, the myosin inhibitor blebbistatin affected only the distribution of MLC (*Figure 5B*, *Figure 5—figure supplement 1B*), and not the phosphorylation of the residues recognized by the antibody (*Ponsaerts et al., 2008*; *Calaminus et al., 2007*). In addition, the overall signaling to p-MLC in cells plated on anti-ICAM-1 antibodies was similar to that of cells on FN (*Figure 5—figure supplement 1C*), indicating that the impaired morphogenesis was specific to ICAM-1 immobilization, and independent of the general actin remodeling and myosin-II activation occurring in response to the adhesion and spreading of cells previously in suspension. Indeed, ICAM-1 surface engagement by clustering with a secondary antibody, independently of epithelial adhesion and spreading, was sufficient to induce the phosphorylation of MLC (*Figure 5E*), which demonstrates that the receptor signals to actomyosin in these cells.

ICAM-1 is the main adhesion receptor for T cells in the hepatic epithelium (*Reglero-Real et al., 2014*). To investigate the relevance of our findings in the context of leukocyte interaction, we analyzed the effect of T-cell adhesion on the apicobasal polarity of hepatic epithelial cells. First, ICAM-1_KO cells had also impaired their ability to interact with T cells, confirming the essential role of epithelial ICAM-1 in T-cell adhesion (*Figure 5—figure supplement 1D*). In addition, co-culture of T-lymphocytes with polarized HepG2 cells for at least 6 hr, at an approximate ratio of 1:2, reduced the number of BCs found in cells prior to adhesion by approximately 40%. Hence, these results indicate that the signaling events that control epithelial apicobasal polarity by ICAM-1 in the absence of interaction with immune cells also mediate the reduction in epithelial polarity and BC frequency upon leukocyte adhesion (*Figure 5F*). Collectively, these results indicate that hepatic ICAM-1 signaling to actomyosin controls canalicular morphogenesis and frequency (*Figure 5G*).

## The proximal interactome of ICAM-1 regulates BC size and ICAM-1-mediated signaling to actomyosin

To characterize further the mechanisms of ICAM-1-mediated signaling to canalicular actomyosin, we analyzed the proteome in the proximity of ICAM-1 by BioID (*Roux et al., 2012*). We expressed a chimeric ICAM-1-BirA* protein, performed a biotinylation assay, and analyzed the neutravidin pull-down fractions by mass

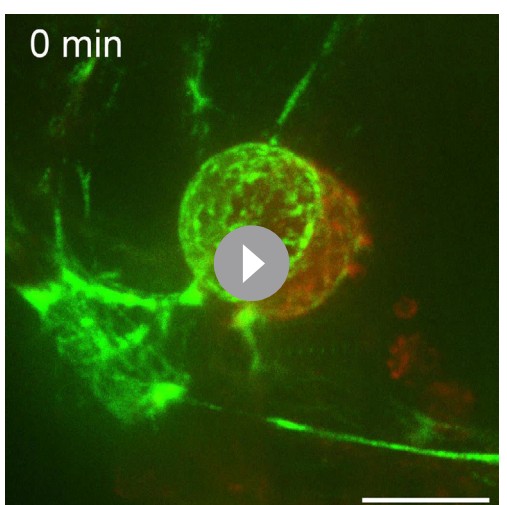

**Video 3.** Time-lapse spinning disc confocal microscopy of polarized HepG2 cells expressing GFP-tagged myosin light chain (GFP-MLC) (green) and incubated with SirActin (red). Images were acquired at 4 min intervals for 48 min and displayed at one frame per second. Scale bar, 3 µm.

https://elifesciences.org/articles/89261/figures#video3

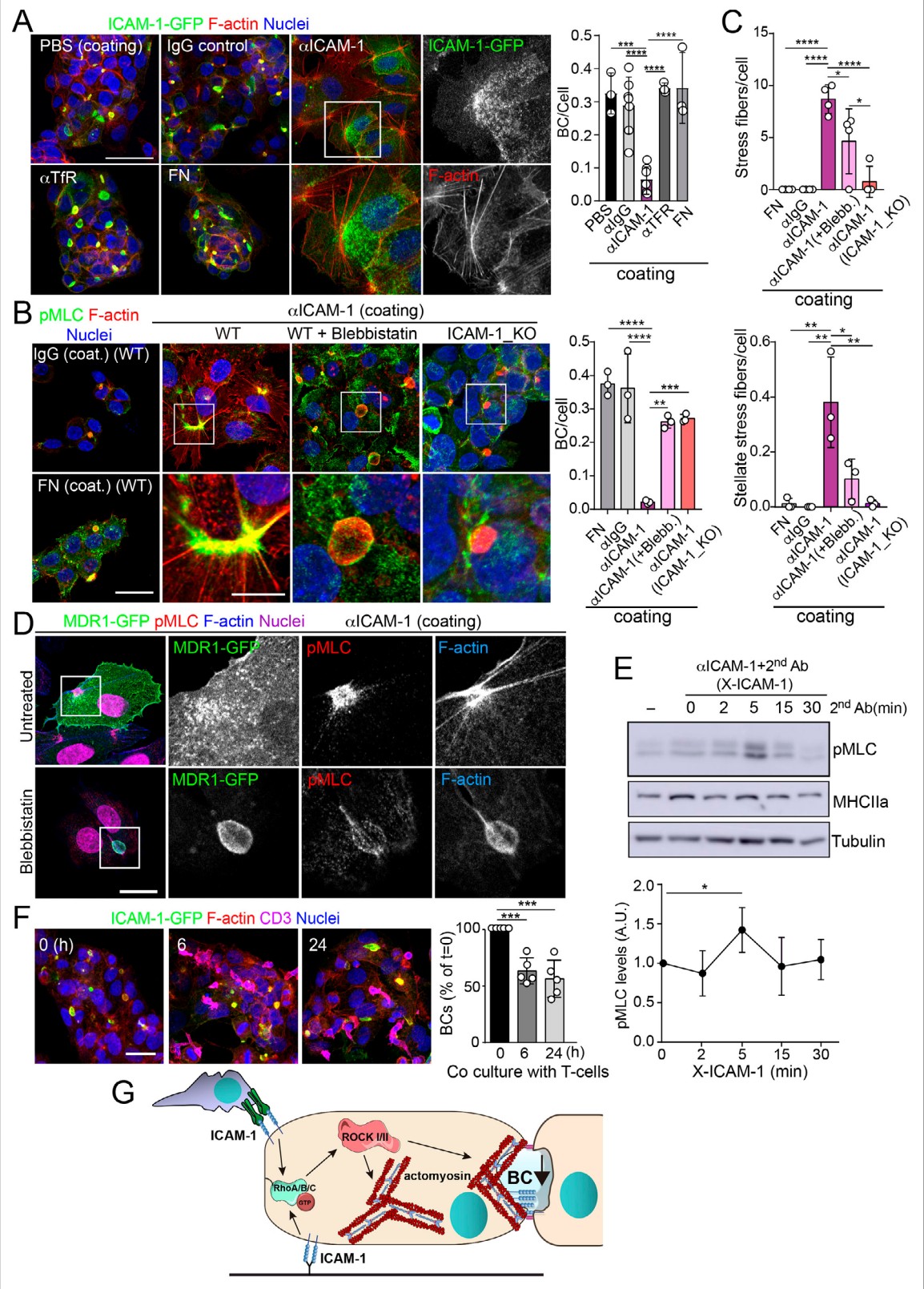

**Figure 5.** Intercellular adhesion molecule-1 (ICAM-1) signals to actomyosin, thereby reducing apicobasal polarity. (**A**) Surface ICAM-1 engagement reduces bile canaliculi (BC) morphogenesis. ICAM-1-GFP HepG2 cells were cultured for 24 hr on coverslips precoated with the indicated antibodies (α), IgG control or fibronectin (FN). Cells were fixed and stained for F-actin to quantify the amount of BCs per cell (right). Nuclei were stained with DAPI. Bars represent the mean ± SD. n = 3. Scale bar, 50 μm. TfR, transferrin receptor. (**B**) Impairment of BC morphogenesis depends on myosin-II.

*Figure 5 continued on next page*

*Figure 5 continued*

ICAM-1-GFP cells were cultured for 24 hr as in (**A**) on coverslips precoated with the indicated proteins. Cells were seeded for 4 hr and incubated with blebbistatin when indicated (blebb.). As a control, ICAM-1_KO cells were plated in parallel on ICAM-1. Cells were fixed, stained for F-actin and pMLC, and the amount of BCs per cell was quantified (right graph). Nuclei were stained with DAPI. Bottom images are enlargements of the boxed areas in the top images showing cells plated on αICAM-1 coverslips. Bars represent the mean ± SD. n = 3. Scale bar, 20 µm; enlarged boxed area 10 µm. Note that blebbistatin inhibits myosin but does not reduce pMLC levels, as previously described. Scale bar, 10 µm. (**C**) Quantification of the stress fibers (top) and stellate stress fibers (bottom) in cells plated on the coverslips precoated with the indicated proteins as in (**B**). n = 4. (**D**) Stellate stress fibers concentrate F-actin and pMLC but not the canalicular marker MDR1. Hepatic epithelial cells expressing MDR1-GFP were plated on coverslips precoated with FN or anti-ICAM-1 antibody (αICAM-1). Scale bar, 15 µm. (**E**) Effect of ICAM-1 clustering (X-ICAM-1) on MLC phosphorylation. Cells were cultured for 24 hr on plastic dishes and then were sequentially incubated with anti-ICAM-1 antibody (first Ab) for 30 min and a specific secondary antibody (second) for the indicated times. Cells were lysed and proteins were detected by western blot. Bottom plot shows the quantification of pMLC levels upon receptor clustering. (**F**) Hepatic epithelial cells stably expressing ICAM-1-GFP were cultured for 72 hr and then incubated with T-lymphocytes (one T cell: two hepatic cells) for the indicated periods. The plot shows the percentage of BCs quantified prior to exposure to T-lymphocytes (0 hr). Bars represent the mean ± SD. n = 5. *p<0.05, **p<0.01, ***p<0.001, ****p<0.0001. Scale bar, 20 µm. (**G**) Current model for the effect of ICAM-1 signaling on hepatic epithelial cells. ICAM-1 engagement signals through the Rho-ROCK axe and increases the formation of actomyosin fibers that prevent the formation of actomyosin-regulated BC structures. In polarized hepatic cells, leukocyte-mediated engagement of ICAM-1 activates pericanalicular actomyosin, which induces contraction at BCs, thereby reducing their frequency.

The online version of this article includes the following source data and figure supplement(s) for figure 5:

**Source data 1.** Original file for the western blot analysis in *Figure 5E* (anti-pMLC).

**Source data 2.** Original file for the western blot analysis in *Figure 5E* (anti-MLC).

**Source data 3.** Original file for the western blot analysis in *Figure 5E* (anti-tubulin).

**Source data 4.** PDF containing *Figure 5E* and original scans of the relevant western blot analysis (anti-pMLC, anti-MLC, and anti-tubulin), with the highlighted bands squared.

**Figure supplement 1.** Localization of ICAM-1 in membrane regions close to stellate stress fibers.

**Figure supplement 1—source data 1.** Original file for the western blot analysis in *Figure 5—figure supplement 1C* (anti-pMLC).

**Figure supplement 1—source data 2.** Original file for the western blot analysis in *Figure 5—figure supplement 1C* (anti-MHCIIa and anti-tubulin).

**Figure supplement 1—source data 3.** Original file for the western blot analysis in *Figure 5—figure supplement 1C* (anti-MHCIIa and anti-tubulin).

**Figure supplement 1—source data 4.** PDF containing *Figure 5—figure supplement 1C* and original scans of the relevant western blot analysis (anti-pMLC, anti-pMHCIIa, and anti-tubulin), with the highlighted bands squared.

spectrometry. Two independent experiments revealed a restricted set of proteins that were biotinylated in both assays and only in cells expressing the ICAM-1-BirA* chimera. Amongst them, the ezrin-radixin-moesin phosphoprotein (EBP)-50/NHERF1/SLC9A3R1 was detected and further validated (*Figure 6A*, *Figure 6—figure supplement 1A–C*). EBP50 is a scaffolding protein that interacts with Rho and Rab GTPase activity regulators (*Hsu et al., 2010*; *Reczek and Bretscher, 2001*) and can bind directly to transmembrane proteins or indirectly through the interaction with ERM proteins, which in turn also bind to Ig-superfamily receptors (*Barreiro et al., 2002*; *Reglero-Real et al., 2014*). However, no proximal biotinylation was detected for ERM proteins (*Figure 6—figure supplement 1D*). EBP50 has been found to interact with ICAM-1 in whole rat hepatic tissue and regulate the liver inflammatory response after bile duct ligation, although the precise molecular mechanisms underlying such regulation, and the cellular identity and localization where such interaction takes place, have not been addressed (*Li et al., 2015*). We found that EBP50 was highly polarized and concentrated and colocalized with ICAM-1 in BCs (*Figure 6B*). Confocal analysis of hepatic tissue also revealed a sinusoidal (S) and canalicular (red arrows) distribution of EBP50 in vivo (*Figure 6C*). However, triple co-staining revealed that EBP50 was exclusively localized in the actin-positive canalicular microvilli but not in the peripheral actomyosin mesh (*Figure 6D*). EBP50 is a modular protein that is maintained in a closed conformation by a head-to-tail intramolecular interaction that masks its association with ligands through its two N-terminal PDZ domains (*Morales et al., 2007*). Immunoprecipitation experiments showed a weak interaction between ICAM-1 and ectopically expressed full-length EBP50, which was strongly increased when the two N-terminal PDZ domains were expressed alone, without the masking C-terminal EB domain (*Figure 6E*). The binding of EBP50 fragments containing only one PDZ domain was clearly reduced and suggested a preferential interaction with the second PDZ domain. The expression of these PDZ domains was sufficient to target EBP50 to BCs (*Figure 6—figure supplement 1E*). Thus, ICAM-1 interacts through this PDZ tandem in addition to a potential indirect interaction through ERM proteins. This latter interaction, which occurs through

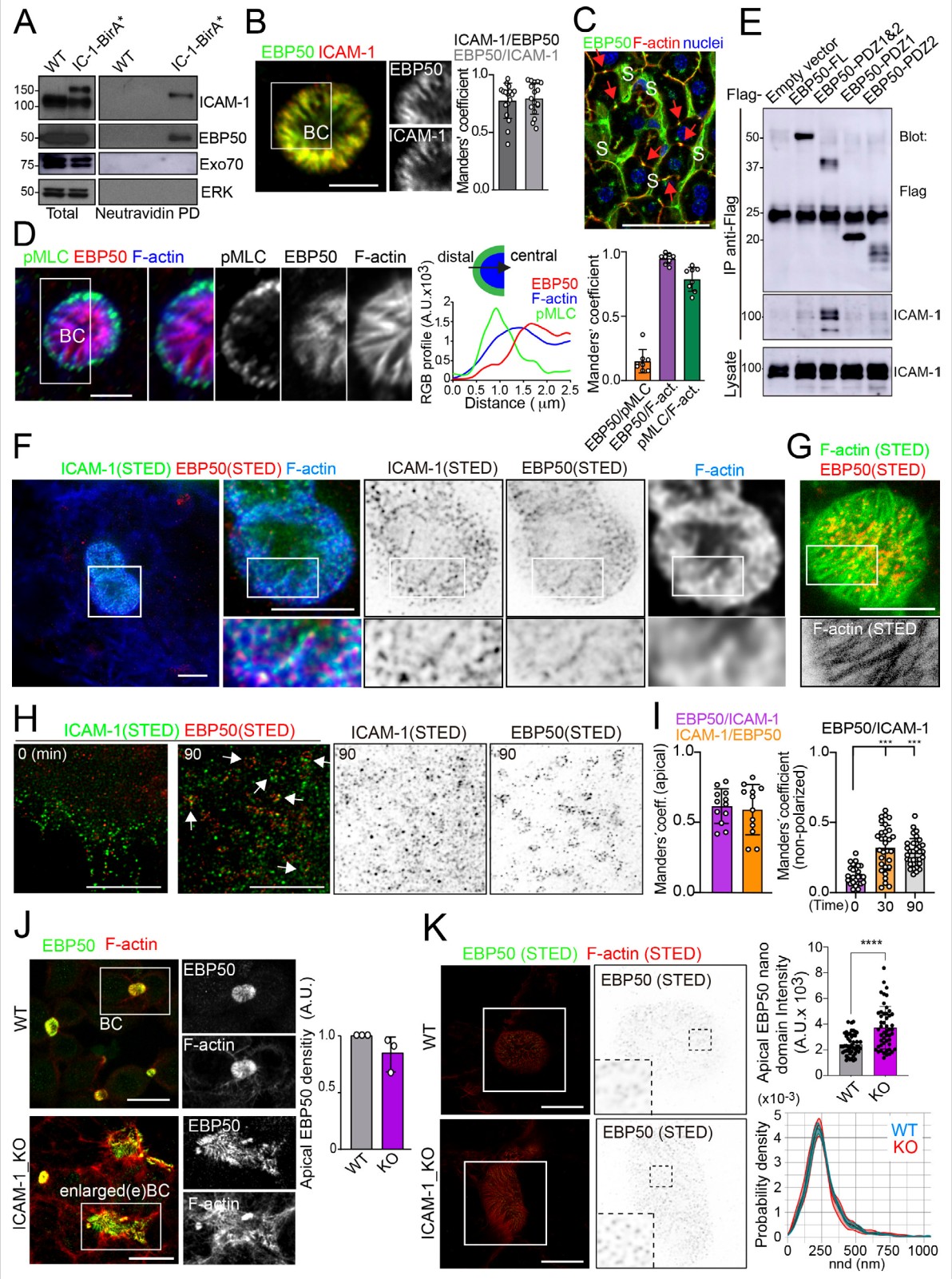

**Figure 6.** Proximal interaction of intercellular adhesion molecule-1 (ICAM-1) with EBP50/NHERF1/SLC9A3R1 into nano-scale microvillar domains. (**A**) The ICAM-1 BioID interactome reveals the proximal interaction of ICAM-1 with EBP50. Parental polarized HepG2 cells (WT) and HepG2 cells stably expressing ICAM-1-BirA* were incubated with 50 µM biotin for 16 hr, lysed and subjected to a pull-down (PD) assay with neutravidin-agarose. Western blots show ICAM-1 and EBP50 biotinylation. Exo70 and ERK are shown as negative controls. ICAM-1 and ICAM-1-BirA* were detected with anti-ICAM-1

*Figure 6 continued on next page*

*Figure 6 continued*

antibody. (**B**) Polarized hepatic epithelial cells were fixed and stained for ICAM-1 and EBP50. BC, bile canaliculi. Manders' analyses for the indicated pairs of staining show the remarkable co-localization between ICAM-1 and EBP50. Scale bar, 5 µm. (**C**) EBP50 is localized in murine hepatic sinusoids (S) and in the liver bile canaliculi (red arrows). Scale bar, 50 µm. (**D**) Triple staining of EBP50, F-actin, and pMLC in BCs. Central panel: relative distribution of the staining intensity for the indicated proteins, starting from the distal parts of BCs, which are enriched in actomyosin, toward the centers of the BCs, which contain the microvillar tips. Right panel: Manders' analyses for the indicated pairs of staining show the remarkable co-localization of EBP50 with F-actin, but not with pMLC. Scale bar, 3 µm. (**E**) Hepatic epithelial cells were transfected with the indicated expression vectors coding for FLAG-tagged full-length EBP50 or its N-termina PDZ domains. 24 hr post-transfection, cells were lysed, subjected to immunoprecipitation with anti-FLAG antibodies, and the immunoprecipitates and the lysates were analyzed by western blot for the indicated proteins. (**F, G**) Polarized hepatic epithelial cells were stained for the indicated proteins and analyzed by super-resolution stimulated emission depletion (STED) super-resolution microscopy. Only two different fluorophores could be subjected to simultaneous STED. (**F**) STED analysis of ICAM-1 and EBP50. Central and right images are a 2.5-fold enlargement of the boxed area in the left image, which corresponds to BCs. Bottom images are a twofold enlargement of the boxed area in top central and right images, which corresponds to a canalicular microvilli-rich area. (**G**) STED analysis of F-actin and EBP50 in a BC. Bottom images are a twofold enlargement of the boxed area in the top image, which corresponds to a canalicular microvilli-rich area. Scale bars, 5 µm. (**H**) STED analyses of non-polarized epithelial cells exposed to anti-ICAM-1 for 30 min in the cold and then incubated at 37°C for the indicated times. ICAM-1 and EBP50 did not overlap at t = 0 in these non-canalicular regions. However, 90 min at 37°C with anti-ICAM-1 antibody induced a redistribution of EBP50 into ring-shaped macroclusters that overlapped with ICAM-1 aggregates. Scale bars, 5 µm. (**I**) Manders' analyses for the indicated pairs of staining from (**F**) (apical) and (**H**) (non-polarized). (**J**) Polarized WT and ICAM-1_KO cells were fixed, stained, and the intensity of EBP50 at the apical BCs was quantified by confocal microscopy (right). Scale bars, 20 µm (**K**) Polarized WT and ICAM-1_KO cells were fixed, stained, and analyzed by STED confocal microscopy. Intensity and nearest-neighbor distance (nnd) of detected spots at BCs and enlarged BCs (eBCs) were calculated. Scale bars, 2 µm. ***p<0.001, ****p<0.0001.

The online version of this article includes the following source data and figure supplement(s) for figure 6:

**Source data 1.** Original file for the western blot analysis in *Figure 6A* (anti-EBP50).

**Source data 2.** Original file for the western blot analysis in *Figure 6A* (anti-EXO70).

**Source data 3.** Original file for the western blot analysis in *Figure 6A* (anti-ICAM-1).

**Source data 4.** Original file for the western blot analysis in *Figure 6A* (anti-ERK).

**Source data 5.** PDF containing *Figure 6A* and original scans of the relevant western blot analysis (anti-ICAM-1, anti-EBP50, anti-EXO70, and anti-ERK), with the highlighted bands squared.

**Source data 6.** Original file for the western blot analysis in *Figure 6E* (anti-ICAM-1 and anti-flag).

**Source data 7.** PDF containing *Figure 6E* and original scans of the relevant western blot analysis (anti-ICAM-1 and anti-flag), with the highlighted bands squared.

**Figure supplement 1.** Intercellular adhesion molecule-1 (ICAM-1) BioID reveals the proximal interaction of the receptor with a new set of proteins.

**Figure supplement 1—source data 1.** Original file for the western blot analysis in *Figure 6—figure supplement 1B* (neutravidin-HRP).

**Figure supplement 1—source data 2.** PDF containing *Figure 6—figure supplement 1B* and original scan of the relevant western blot (neutravidin-HRP), with the highlighted region squared.

**Figure supplement 1—source data 3.** Original file for the western blot analysis in *Figure 6—figure supplement 1D* (anti-ICAM-1 and anti-SNAP23).

**Figure supplement 1—source data 4.** Original file for the western blot analysis in *Figure 6—figure supplement 1D* (anti-ERMs).

**Figure supplement 1—source data 5.** PDF containing *Figure 6—figure supplement 1D* and original scan of the relevant western blot (anti-ICAM-1, anti-ERMs, and anti-SNAP23), with the highlighted region squared.

**Figure supplement 1—source data 6.** Original file for the western blot analysis in *Figure 6—figure supplement 1F* (anti-ICAM-1).

**Figure supplement 1—source data 7.** Original file for the western blot analysis in *Figure 6—figure supplement 1F* (anti-EBP50).

**Figure supplement 1—source data 8.** Original file for the western blot analysis in *Figure 6—figure supplement 1F* (anti-tubulin).

**Figure supplement 1—source data 9.** Original file for the western blot analysis in *Figure 6—figure supplement 1F* (anti-ERMs).

**Figure supplement 1—source data 10.** Original file for the western blot analysis in *Figure 6—figure supplement 1F* (anti-GAPDH).

**Figure supplement 1—source data 11.** PDF containing *Figure 6—figure supplement 1F* and original scan of the relevant western blot (anti-ICAM-1, anti-EBP50, anti-tubulin, anti-ERMs, and anti-GAPDH), with the highlighted region squared.

the EB domain of EBP50, could not be properly analyzed because of the very low levels of ectopic expression obtained for this fragment alone, which suggests that the EB domain is unstable and prone to degradation in the absence of the rest of the protein (not shown). To better understand the ICAM-1-EBP50 molecular complexes, we analyzed the canalicular distribution of these two proteins by super-resolution stimulated emission depletion (STED) confocal microscopy (*Figure 6F*). STED revealed that both ICAM-1 and EBP50 distribution was not even, but displayed patterns of over-lapping nano-scale domains along F-actin-rich microvilli. These nanodomains had a size between 80

and 200 nm. In contrast, super-resolution images showed that microvillar F-actin did not form nanoscopic clusters in these filaments (*Figure 6G*). Furthermore, in plasma membrane regions containing few microvilli, such as those of non-polarized cells, ICAM-1 was more evenly distributed and did not overlap with EBP50 (*Figure 6H*). These EBP50 nanodomains, however, responded to ICAM-1 engagement by aggregating themselves, often forming circular structures of several nanoclusters, which only partially overlapped with the clustered receptor (*Figure 6H and I*). Such redistribution caused EBP50 to become more insoluble to mild-non-ionic detergents (*Figure 6—figure supplement 1F*), indicating that ICAM-1 induces changes in the condensation properties of EBP50, either by increasing the association of EBP50 to cytoskeletal structures or by changing its oligomerization properties (*Fouassier et al., 2000*). Importantly, ICAM-1 and EBP50 nanoclusters were in closer proximity in apical BCs than in non-polarized plasma membrane regions, even if non-polarized cells were exposed to anti-ICAM-1 antibody, which increased nanocluster overlapping, as shown by Manders' analyses (*Figure 6I*). This indicates that the proximal interaction and the potential ICAM-1-EBP50-mediated signaling mainly occur in BCs in the absence of receptor engagement. In addition, it also suggests that, in addition to associating to this scaffold protein, ICAM-1 also signals and reorganizes EBP50 in macromolecular complexes that do not contain the receptor.

The absence of ICAM-1 in KO cells caused no overall dispersion of EBP50 from BCs, indicating that EBP50 is localized in these apical plasma membrane domains through additional interactions (*Figure 6J*). Nevertheless, quantification of EBP50 by STED microscopy revealed an increase in the intensity of the nanodomains in ICAM-1_KO cells with respect to WT, but not in their distribution and density, measured by analyzing the nearest-neighbor distance (*Figure 6K*, nnd). These observations suggest that EBP50 nanoclustering does not require ICAM-1 expression, but also that the receptor modulates EBP50 clustering at BCs. Next, we addressed whether EBP50 regulates canalicular size and dynamics. siRNA-mediated *SLC9A3R1* gene silencing reduced EBP50 expression by around 50% and increased BC area by almost half (*Figure 7A and B*). Although moderately, this EBP50 depletion decreased pMLC staining at the canalicular actomyosin network. Unlike ICAM-1_KO cells, F-actin levels at the canalicular microvilli of EBP50_KD cells were lower than that in control cells, which suggests that EBP50 has functions that are independent of ICAM-1 in these apical domains (*Figure 7B*). It is particularly noteworthy that EBP50 depletion attenuated the negative effect of ICAM-1 on BC morphogenesis and ICAM-1-mediated signaling to F-actin (*Figure 7C–E*). EBP50_KD cells formed more BCs upon ICAM-1 immobilization and reduced the number of stress fibers with respect to control cells (*Figure 7E*). In addition, EBP50 accumulated with ICAM-1 close to stellate stress fibers, consistent with the interaction between ICAM-1 and EBP50 in membrane domains enriched in underlying F-actin (*Figure 7D*, top images). These results indicate that EBP50 regulates ICAM-1-mediated signaling toward actomyosin. Such signaling prevents BC morphogenesis and can promote BC contraction and loss of apicobasal polarity. Given the role of ICAM-1 association with EBP50 in experimental cholestatic liver injury (*Li et al., 2015*), our findings provide a mechanistic rationale to explain the function of these molecular complex in mediating the progression of inflammatory diseases that disrupt the biliary network.

## ICAM-1 regulates the size and morphology of canalicular-like structures in an actomyosin-dependent manner in hepatic organoids

A confocal analysis of ICAM-1 distribution in murine livers revealed that this receptor was highly expressed in hepatic sinusoids and, with less intensity, in bile canaliculi and ducts (*Figure 8A*; *Colás-Algora et al., 2020*). Hence, we performed a morphological study of BC in livers from ICAM-1_KO mice and their WT littermates (*Bullard et al., 2007*), which revealed a significant increase in canalicular width in hepatic tissue lacking ICAM-1 expression (*Figure 8B*). This suggested that ICAM-1 regulates the biliary network of murine livers. Canalicular components such as F-actin and MRP2 were still detected in BC from ICAM-1_KO livers, consistently with the results observed in human hepatic cells (*Figure 8C*). However, the specific cell-autonomous contribution of epithelial ICAM-1 with respect to the sinusoidal expression of the receptor and, above all, with respect to the adhesion of immune cells to hepatic epithelial and endothelial ICAM-1-expressing cells could not be discriminated by analyzing these tissues. To investigate the potential cell-autonomous role of epithelial hepatic ICAM-1 on regulating canalicular membrane domains and cell polarity, independent of other functions in vessels and of its interactions with leukocytes, we took advantage of recent advances in the generation of hepatic

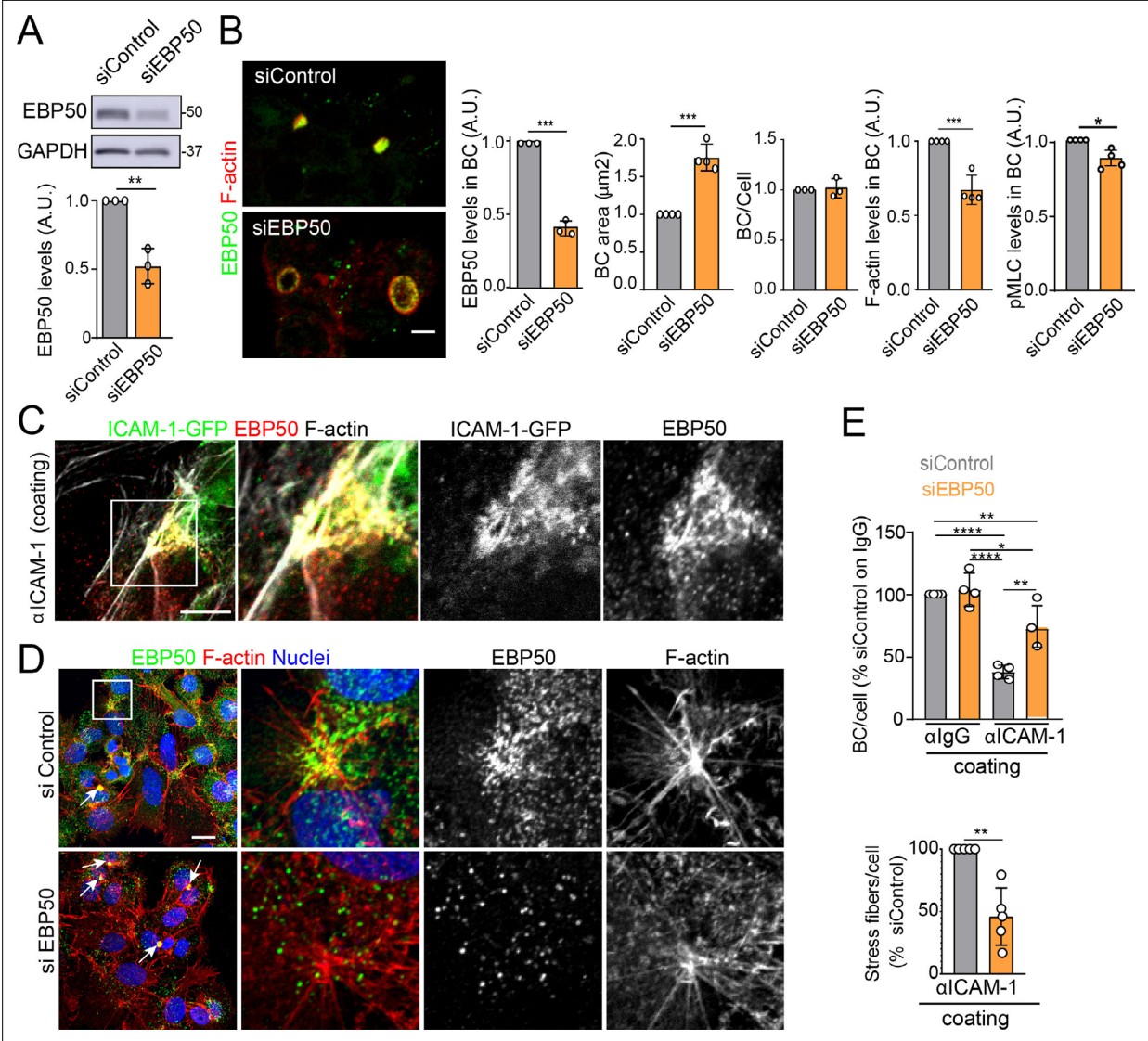

**Figure 7.** EBP50 regulates intercellular adhesion molecule-1 (ICAM-1)-mediated signaling. (**A, B**) HepG2 cells were transfected with siRNA control (siControl) or with siRNA targeting EBP50 (siEBP50) and cultured for 72 hr. Cells were lysed and analyzed by western blot for the indicated antibodies (**A**) or fixed and stained for the indicated proteins (**B**) Bile canaliculi (BC) frequency and area and the intensity levels of the indicated protein at BCs were quantified. Bars represent the mean ± SD of at least three experiments. Scale bar, 10 µm. (**C**) HepG2 cells stably expressing ICAM-1-GFP were cultured for 24 hr on coverslips precoated with anti-ICAM-1 antibody, fixed and stained for F-actin and EBP50. Central and right images are an enlargement of the boxed area in the left confocal image, which shows a region enriched in stress fibers and stellate stress fibers. Scale bar, 5 µm. (**D**) HepG2 cells were transfected with siControl or siEBP50 for 48 hr, then cultured for 24 hr on coverslips precoated with anti-ICAM-1 antibody. Cells were fixed and stained for F-actin and EBP50. Central and right images are enlargements of the boxed areas in the left images, which show regions enriched in stress fibers and stellate stress fibers. Arrows point at BCs. Scale bar, 10 µm. (**E**) Quantification of the number of BC per cell (top) and the percentage of stress fibers per cell with respect to siControl-transfected cells (bottom) of experiments shown in (i). Bars represent the mean ± SD of at least four experiments. *p<0.05, **p<0.01, ***p<0.001, ****p<0.0001. A.U., arbitrary units. Nuclei were stained with DAPI.

The online version of this article includes the following source data for figure 7:

**Source data 1.** Original file for the western blot analysis in *Figure 7A* (anti-EBP50 and anti-GAPDH).

**Source data 2.** PDF containing *Figure 7A* and original scans of the relevant western blot analysis (anti-EBP50 and anti-GAPDH), with the highlighted bands squared.

organoids obtained from 3D culture and expansion of ductal bipotent stem cells isolated from human and murine livers (*Figure 8—figure supplement 1A*; *Broutier et al., 2016*; *Huch et al., 2015*). Murine hepatic organoids expressed hepatocyte markers such as albumin. They also expressed ICAM-1 without previous exposure to inflammatory stimuli (*Figure 8—figure supplement 1B*; *Broutier et al.,*

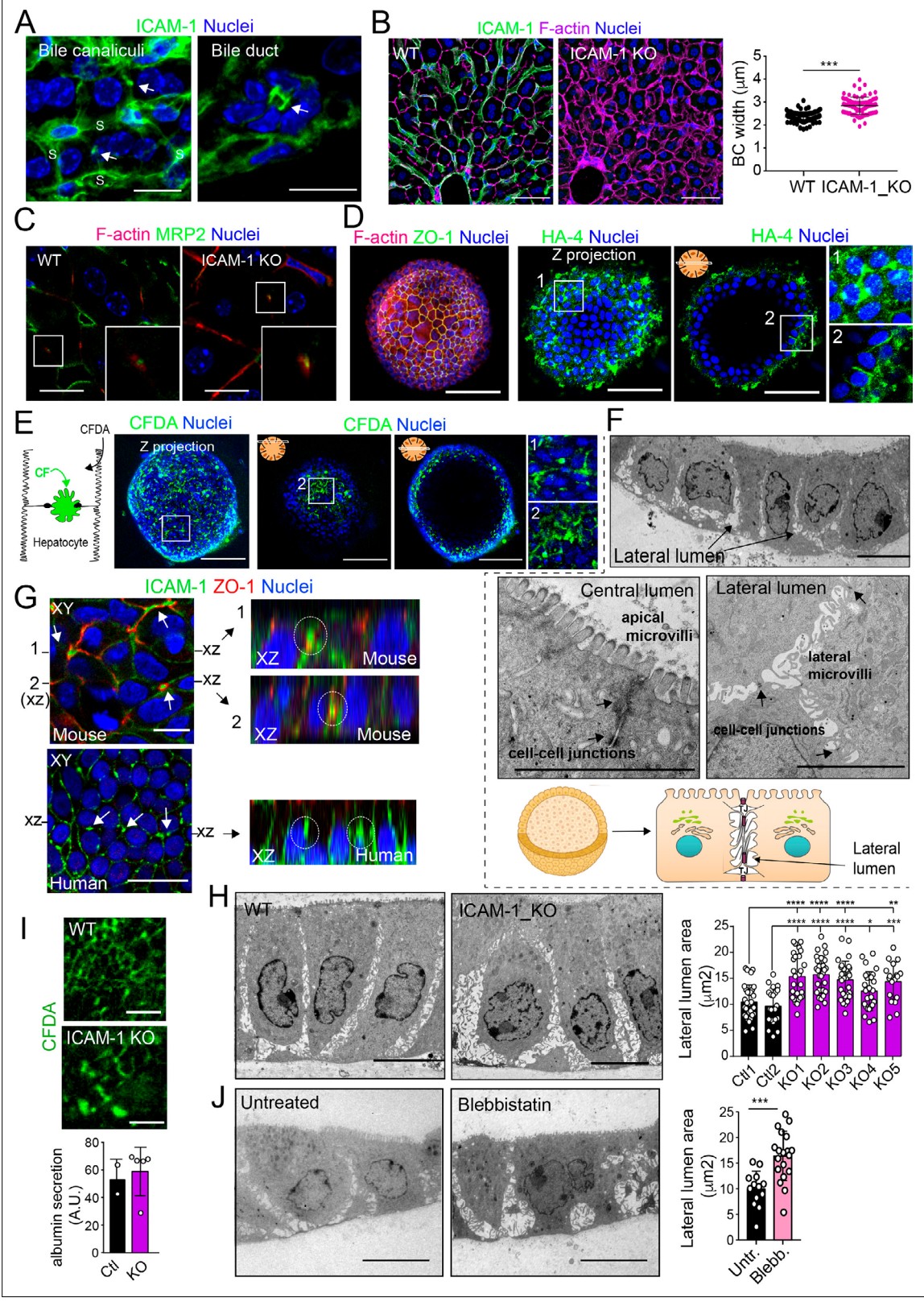

**Figure 8.** Intercellular adhesion molecule-1 (ICAM-1) and non-muscle myosin-II regulate the size of canalicular-like cavities in hepatic organoids. (**A**) ICAM-1 concentrates in bile canaliculi and bile ducts in livers of LPS-stimulated mice (arrows). S, sinusoids. Scale bars, 20 µm (left image) and 10 µm (right image). (**B**) ICAM-1, F-actin, and nucleus staining of livers from WT and *Icam1*_KO mice. Scale bars, 20 µm. Right graph: quantification of bile canaliculi (BC) width from WT and ICAM-1_KO murine livers identified morphologically by F-actin staining more than 80 bile canaliculi from four WT and

*Figure 8 continued on next page*

*Figure 8 continued*

four *Icam1*_KO mice. (**C**) Immunohistochemical analyses of F-actin and MRP2 in bile canaliculi in WT and *Icam1*_KO mice. Insets show enlargements of the boxed areas. Scale bars, 6 μm. (**D**) Mouse liver organoids have a spherical morphology. Differentiated hepatic organoids were fixed and stained for the indicated antibodies and for F-actin and nuclei to visualize cell morphology. A Z-stack projection of an organoid is shown on the left central image. A single confocal image is shown on the central right image. Enlargement of the boxed areas is shown on the right. Scale bars, 100 μm. (**E**) Mouse liver organoids were treated with 25 μM CFDA for 30 min. Left panel shows the Z-stack projections of different confocal planes of an organoid. Central and right panels show the distal and central confocal planes, respectively, of the organoid in the left images. Right images show a twofold magnification of the boxed areas. Scale bars, 100 μm. (**F**) Mouse liver organoids were fixed and processed for transmission electron microscopy. Cells organized in sheets facing big central lumens (central left image) but also formed cavities between their lateral membranes containing cell–cell junctions and microvilli (lateral lumens) (top and central right images, bottom cartoon). Scale bars, 5 μm. (**G**) Mouse (top) and human (bottom) liver organoids were fixed, permeabilized, and stained for ICAM-1, F-actin, and nuclei. Left images show a confocal plane that crosses the central part of a cell sheet. F-actin and ICAM-1 accumulations were detected between cells (arrows). ZO-1 staining clearly surrounded these accumulations in the mouse organoid (bottom). Black lines on the side of the left images mark the location of the X-Z reconstructions shown in the right images. Encircled regions show lateral areas of ICAM-1 accumulation. (**H**) Ultrastructural analysis of differentiated WT and ICAM-1_KO liver organoids. Quantification of lateral lumen areas for organoids from two WT and five *Icam1*_KO mice (right). Areas of more than 20 lateral lumens were analyzed for each mouse. Bars represent the mean ± SD. Scale bars, 5 μm. (**I**) Functional comparison of WT and ICAM-1_KO organoids. WT and ICAM-1_KO organoids equally processed and secreted CFDA into lateral lumens (top images) and secreted albumin (bottom graph). A.U., arbitrary units. (**J**) Ultrastructural analysis of differentiated WT liver organoids treated or not with 10 μM blebbistatin for 24 hr. Scale bars, 5 μm. The right plot shows the quantification of lateral lumen areas. Quantifications represent the mean ± SD. *p<0.05, **p<0.01, ***p<0.001, ****p<0.0001. Nuclei were stained with DAPI in confocal images.

The online version of this article includes the following source data and figure supplement(s) for figure 8:

**Figure supplement 1.** Generation of liver organoids from bipotent precursor cells and the effect of ICAM1 gene knockout on their cellular morphology.

**Figure supplement 1—source data 1.** Original file of the gel containing PCR analyses of several transcripts in murine liver organoids grown in expanding medium and in hepatic tissue shown in *Figure 8—figure supplement 1B* (panels ICAM-1 and hrpt).

**Figure supplement 1—source data 2.** Original file of the gel containing PCR analyses of several transcripts of murine liver organoids cultivated in expanding medium (EM) and differentiated medium (DM) and of liver tissue shown in *Figure 8—figure supplement 1B* (panels ICAM-1 and albumin).

**Figure supplement 1—source data 3.** PDF containing *Figure 8—figure supplement 1B* and original scan of the relevant PCR analyses (ICAM-1, hrpt, albumin), with the highlighted region squared.

*2016*; *Huch et al., 2013*). Hepatic organoids had a heterogeneous morphology but mostly formed spheroids of cell aggregates that formed cell–cell junctions (*Figure 8D*, *Figure 8—figure supplement 1C*). Interestingly, the bile canalicular marker HA-4 was localized between cells in the peripheral regions of these spheroids. HA-4 exhibited a staining pattern of wide tubular and dotted domains different from that of the cell–cell junction marker ZO-1 (*Figure 8D*, enlarged areas). Moreover, a time-lapse analysis of carboxyfluorescein diacetate (CFDA), a fluorescent probe of hepatobiliary transport and secretion into the bile canaliculi, revealed that fluorescent CF was also laterally secreted between cells, as shown by the relative distribution of CF with respect to the nuclei, which was similar to that of HA-4 (*Figure 8D and E*). Transmission electron microscopy of these organoids revealed the existence of lateral cavities between cells, which were sealed by electrodense cell–cell junctions and contained long, thin microvilli, different from those observed in the plasma membrane domain facing bigger lumens in the central part of the spheroids (*Figure 8F*). The cell–cell junctions observed in these lateral lumens resembled the recently described membrane bulkheads that contribute to the elongation of canalicular lumens (*Figure 8F*, bottom-right image) (*Belicova et al., 2021*). Collectively, these results suggested that differentiated hepatic organoids form BC-like lateral cavities that morphologically and functionally resemble BCs. Interestingly, ICAM-1 accumulated in these domains in mouse and human hepatic organoids (*Figure 8G*). pMLC and EBP50 were also localized in lateral plasma membrane domains, although EBP50 also concentrated in the plasma membrane facing central lumens (*Figure 8—figure supplement 1C*). To address whether ICAM-1 regulates the morphology of these BC-like lateral cavities, we generated hepatic organoids from WT and ICAM-1_KO mice. Measuring the area of BC-like lateral lumens revealed larger areas in ICAM-1_KO than that in WT-differentiated organoids (*Figure 8H*). Time-lapse analyses of organoids incubated with CFDA (*Figure 8I*, top) as well as albumin secretion assays (*Figure 8I*, bottom), indicated that BC-like lateral domains were functional in WT and ICAM-1_KO organoids. In addition, confocal (*Figure 8—figure supplement 1D and E*) and electron (*Figure 8—figure supplement 1F*) microscopy analyses also revealed that ICAM-1_KO organoids contained enlarged cells (*Figure 8—figure supplement 1G*), suggesting that ICAM-1 may exert an overall control of the submembranal cytoskeletal scaffolds that control cellular size in the

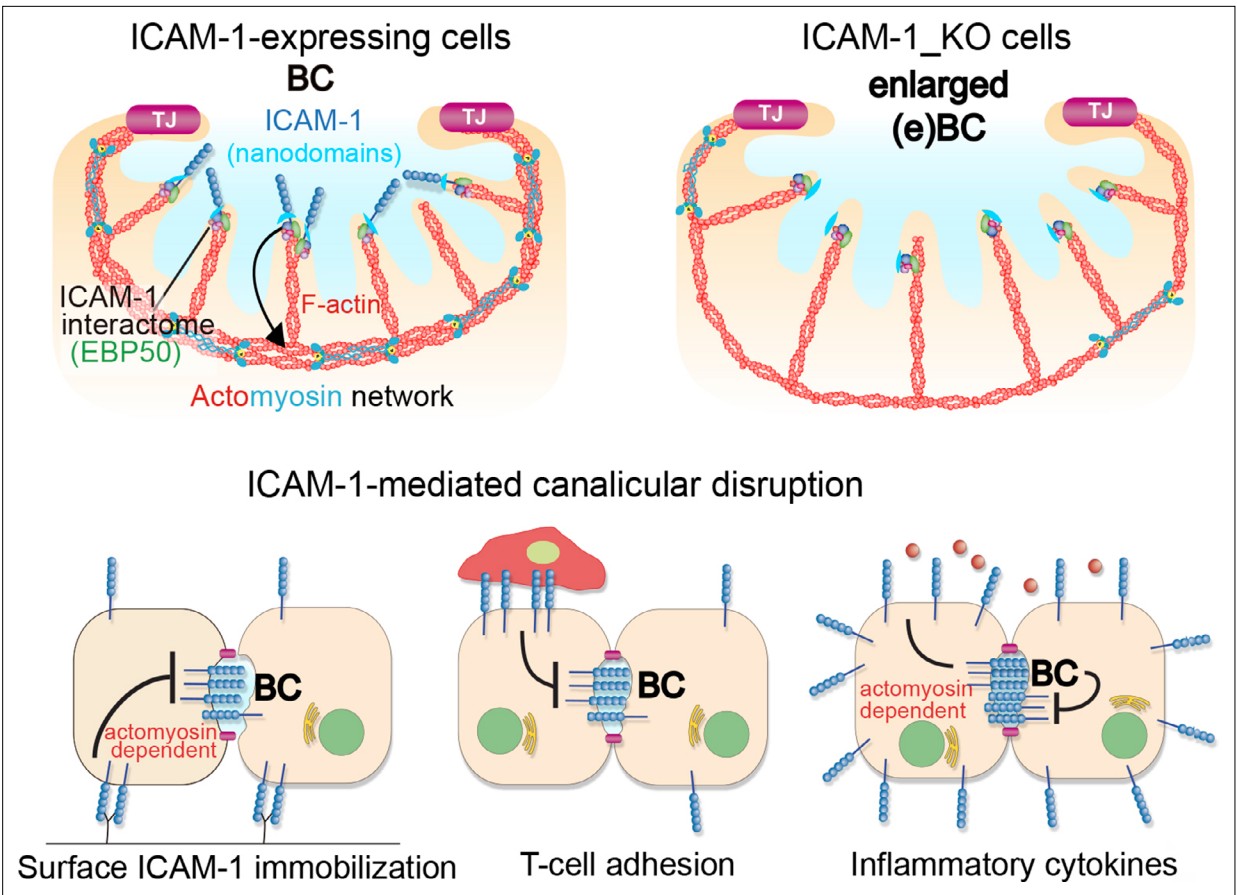

**Figure 9.** A model for the control of actomyosin-mediated contraction of bile canaliculi (BC) mediated by intercellular adhesion molecule-1 (ICAM-1) expression levels, which determine the size and frequency of these apical structures in hepatic epithelial cells. Top images: canalicular ICAM-1 signals toward a distal canalicular actomyosin network at the base of microvilli. EBP50 links ICAM-1 to F-actin and localizes into membrane nano-scale domains to mediate this signaling. In ICAM-1_KO, actomyosin-mediated contraction is reduced and BCs are enlarged (eBCs). Bottom images: ICAM-1 can signal toward actomyosin and regulate BC frequency. Surface immobilization of ICAM-1 prevents BC formation (bottom left) or disrupts mature BCs, as shown by ICAM-1-mediated T-cell adhesion (bottom center). ICAM-1 upregulation in response to inflammatory cytokines increases actomyosin-mediated contraction at BCs, which also reduces BC frequency (bottom right).

hepatic organoids. Finally, incubation with blebbistatin also increased the area of these lateral lumens in a manner almost identical to that observed in ICAM-1_KO organoids (*Figure 8J*).

Hence, collectively, our observations from the various polarized hepatic cellular models reveal that ICAM-1 is organized in nano-scale domains in surface microvilli at BCs, from where the receptor signals to actomyosin, independent of its role as an adhesion receptor for immune cells. This apical confinement of ICAM-1 regulates BC size and the hepatic cell architecture. In a more general perspective and taking into account previous reports (*Reglero-Real et al., 2014*; *Cacho-Navas et al., 2022*), these findings also provide direct evidence of a close and reciprocal connection between epithelial apicobasal polarity and the inflammatory response through ICAM-1 (*Figure 9*).

## Discussion

We previously showed that epithelial apicobasal polarity modulates leukocyte adhesion to hepatic epithelial cells by regulating ICAM-1 exposure to the extracellular milieu (*Reglero-Real et al., 2014*). Our findings here indicate that ICAM-1-mediated signaling regulates epithelial cell polarization. This signaling can emanate from receptor engagement upon leukocyte adhesion, in accordance with previous reports in different cellular contexts (*Millán and Ridley, 2005*; *Etienne et al., 1998*), but also by confining and enriching ICAM-1 in BCs in a leukocyte-independent manner. The participation of ICAM-1 in immune cell adhesion and trafficking through the vascular system, which are

essential immune and inflammatory processes closely involved in liver pathology, makes it difficult to discriminate and quantify in vivo the real contribution of epithelial ICAM-1 to the loss of epithelial architecture during the progression of these pro-inflammatory diseases, independently of its function in leukocyte adhesion. However, our experiments with different cellular models demonstrate that ICAM-1 upregulation upon proinflammatory stimulation is sufficient to depolarize hepatic epithelial cells in an actomyosin-dependent manner. The concentration of this receptor in nano-scale domains in membrane regions enriched in microvilli, such as BCs, is sufficient to induce signaling to actomyosin, which regulates the contraction of these apical structures, leading to their disruption. Thus, given that many proinflammatory liver diseases, which provoke massive leukocyte infiltration into the liver, cause a reduction in epithelial apicobasal polarity in the hepatic parenchyma (*Gissen and Arias, 2015*), leukocyte-dependent and independent-ICAM-1 signaling may directly contribute to the pathological dysfunction of hepatocytes and cholangiocytes.

Although ICAM-1 localizes to and signals from the basolateral membrane, polarized hepatic cells express basolateral-to-apical transcytotic machinery that removes the receptor from these plasma membrane domains and confines most of ICAM-1 within the BC (*Reglero-Real et al., 2014*; *Cacho-Navas et al., 2022*). This machinery is developmentally expressed in other epithelia and is essential for the acquisition of cell polarity and differentiation of gut epithelial cells, highlighting again the molecular connections between epithelial patterning and immune response. By editing and silencing the *ICAM1* gene, or by increasing its expression in response to inflammatory cytokines, we have shown that this apical confinement controls the canalicular dynamics by regulating the pericanalicular myosin-II levels. Bile canaliculi are mechanosensory hubs that polarize actomyosin upon changes in canalicular flow and during the regeneration of the hepatic parenchyma (*Meyer et al., 2020*). In accordance with these findings, we have found that BCs concentrate not only F-actin and actomyosin but also their master regulators, the GTPases of the RhoA subfamily. In vivo, myosin-II exhibits a canalicular distribution, although discriminating between subapical and fully apical distributions by high-resolution light microscopy is more difficult to achieve in vivo under some experimental conditions (*Meyer et al., 2020*). However, a detailed analysis of liver samples by electron microscopy has identified a subapical myosin-II network surrounding the microvillus-rich canalicular plasma membrane (*Tsukada et al., 1995*). This myosin-II activity controls the function of the bile canaliculus in vivo (*Meyer et al., 2017*). Pharmacological inhibition of ROCK is sufficient to increase the bile canalicular diameter in murine livers, which reduces the bile flow and delays CFDA clearance from this network of channels (*Meyer et al., 2017*). Conversely, cholestasis-inducing drugs affect hepatic myosin-II activation, suggesting that canalicular actomyosin may be a target when treating diseases related to the bile flow (*Sharanek et al., 2016*). On the other hand, blebbistatin and Y-27632 treatments are also sufficient to induce lateral, canalicular-like lumen formation in columnar epithelial cells when cell–cell junction formation is prevented. This suggests a central and specific role for the RhoA-ROCK-myosin-II pathway in forming lateral canalicular-like structures (*Cohen et al., 2007*). The enlargement of hepatic apical membranes has also been observed in the liver parenchyma in various pathological and physiological contexts, and resembles alterations observed in cells lacking proteins regulating apical membrane integrity, such as ABCB4 and Cdc42 (*Pradhan-Sundd et al., 2020*; *Shitara et al., 2019*; *Sidhaye et al., 2014*).

Myosin-II has been located in the terminal web, the cytoskeletal network at the base of microvilli, in a variety of epithelial cells (*Delacour et al., 2016*). Myosin-II activity in this structure negatively regulates the incorporation of G-actin into microvilli, which affects their length (*Chinowsky et al., 2020*). The myosin-II grid detected in the base of BCs under our experimental conditions displays a discrete pattern that colocalizes with the most basal part of microvilli, suggesting that this cytoskeletal structure is also part of the hepatic epithelial terminal web. Therefore, crosstalk appears to exist between myosin-II at the terminal web and microvilli. If ICAM-1 expression and localization in canalicular microvilli determine the activation of pericanalicular myosin-II that controls canalicular dynamics, it is plausible that the receptor may regulate the canalicular network in proinflammatory diseases in which hepatic ICAM-1 density is massively increased. However, the effect of inflammatory cytokines on the bile flow has not been adequately addressed so far. We know that TNF-α and IL-6 treatments clearly reduce bile flow in rat livers and alter canalicular dynamics in vitro (*Ikeda et al., 2003*). Liver pathologies with an inflammatory component have in common the loss of apicobasal polarity of epithelial cells in the hepatic parenchyma (*Shousha et al., 2004*). Our results stimulating polarized

hepatic cells suggest that cellular depolarization is intrinsic to expression changes occurring in these cells upon exposure to inflammatory cytokines and that ICAM-1 induction contributes, at least in part, to polarity loss without requiring engagement by adhering leukocytes.

ICAM-1 is a type I transmembrane protein that mediates plasma membrane remodeling upon engagement because it is connected to the underlying actin cytoskeleton by membrane-actin cross-linkers such as ERMs. In endothelial cells, engaged ICAM-1 receptor reorganizes microvilli into docking structures that mediate leukocyte adhesion and promote extravasation (*Reglero-Real et al., 2012*; *Millán et al., 2006*; *Barreiro et al., 2002*). Therefore, in other cellular contexts, ICAM-1 may help form plasma membrane domains in which the receptor is also confined. Photoactivation experiments have demonstrated that ICAM-1 concentrates and is highly stable in BCs, whereas in the basolateral plasma membrane the receptor rapidly diffuses and translocates toward the BC (*Reglero-Real et al., 2014*). This suggests that molecular mechanisms must exist that selectively stabilize or engage ICAM-1 in these apical membrane domains. Our interactome analyses revealed that ICAM-1 interacts with the scaffold protein EBP50/NHERF1/SLC9A3R1, which is also confined in canalicular microvilli. Moreover, super-resolution microscopy experiments clearly demonstrate that ICAM-1 and EBP50 are organized in overlapping nanoclusters in BC. The nature of these domains remains to be solved, but F-actin does not form clusters and is evenly distributed in microvilli, suggesting that other nano-scale structures should be involved. Liquid-ordered membrane condensates or lipid rafts help form membrane microdomains based on the differential affinity of certain membrane lipids for membrane-associated proteins (*Levental and Lyman, 2023*), thereby regulating clustering and function of trans-membrane proteins. Lipid-ICAM-1 membrane nanodomains may be involved in the control of ICAM-1 nanodomains and their subsequent signaling since these condensates are abundant in BCs (*Ismair et al., 2009*) and a transient confinement in raft-like domains has been previously reported for ICAM-1 (*Millán et al., 2006*; *Tilghman and Hoover, 2002*) and EBP50 (*Sultan et al., 2013*) in different cellular contexts.

ICAM-1-actin connection through EBP50 or ERM proteins into overlapping nanodomains may therefore have a role in its apical confinement, but additional interactions mediated by ICAM-1 trans-membrane and extracellular domains must contribute to its retention and function in the BCs. Ligand-mediated stabilization may be one of the mechanisms contributing to ICAM-1 retention in BCs. It is unlikely that polarized hepatic cells express and accumulate β2 integrin molecules on the canalicular surface that are able to interact in *trans* with ICAM-1. In addition, hepatic epithelial cells express very low levels of Muc1, which is another ICAM-1 ligand involved in epithelial cancer cell adhesion (*Regimbald et al., 1996*). However, hepatocytes are the main source of fibrinogen, a large protein involved in coagulation, which binds and clusters the ICAM-1 receptor by interacting with its first Ig-like domain (*Tsakadze et al., 2002*). As shown by The Human Protein Atlas (*Uhlén et al., 2015*; *Yu et al., 2015*), hepatic cells express very high levels of various fibrinogen chains and at least the fibrinogen β-chain is highly concentrated in bile canaliculi in immunohistochemical analyses of human hepatic tissue. This suggests that fibrinogen is not only secreted, but also retained close to hepatic apical membrane domains, where it could potentially interact with, cluster, and stabilize canalicular ICAM-1, thereby inducing its intracellular signaling.

In conclusion, our findings indicate that ICAM-1 plays a dual role in hepatic epithelial cells: as an adhesion receptor that interacts with leukocytes and guides immune cell migration, and as a signaling molecule that regulates non-muscle myosin-II through EBP50, independent of its role as a leukocyte adhesion counter receptor, and mostly in BCs (*Figure 9*). Interestingly, our results also bring together molecular machinery that is central to the progression of cholestasis in preclinical models, providing potential mechanistic insights into the canalicular collapse provoked by this pathological condition (*Gupta et al., 2017*; *Li et al., 2015*), which is linked to inflammatory responses that induce ICAM-1 expression (*Gujral et al., 2004*; *Li et al., 2017*). Further investigation into the leukocyte-independent signaling pathways orchestrated by ICAM-1 may reveal new strategies to preserve the hepatic epithelial architecture in different pathological inflammatory contexts.

# Materials and methods

## Lead contact and materials availability

Further information and requests for resources and reagents should be directed to, and will be fulfilled by, the lead contact, Jaime Millán (jmillan@cbm.csic.es).

All unique/stable reagents generated in this study will be made available on request, but we may require a Material Transfer Agreement.

## Cells and culture

Human polarized hepatic HepG2 cells were grown in high-glucose Dulbecco's modified Eagle's medium supplemented with 5% fetal bovine serum (ATCC HB-8065). T lymphoblasts were prepared from isolated human peripheral blood mononuclear cells (PBMCs). Nonadherent PBMCs were stimulated with 0.5% phytohemagglutinin for 48 hr and maintained in RPMI medium supplemented with 2 U/ml IL-2, as previously described (*Millán et al., 2002*). Experiments were performed with memory T-lymphocytes cultured for 7–12 d. Inflammatory stimulation of human polarized hepatic cells was performed with 50 ng/ml TNF-α, 15 ng/ml IL-1β, and IFN-γ 1000 U/ml. These concentrations were identified after performing initial dose–response curves and analyses of ICAM-1 expression and STAT3 and AKT phosphorylation in these cells and in human endothelial cells (*Colás-Algora et al., 2023*).

## Biliary ductal stem cell isolation and organoid culture

Wildtype and *Icam1*_KO mice on a C57BL/6 background were used in accordance with the institutional Animal Welfare Ethical Review Body and UK Home Office guidelines. *Icam1*_KO mice were kindly shared by Prof. Nancy Hogg (London, UK). Briefly, *Icam1*_KO mice were generated by Dr. Arthur L. Beaudet (Baylor College of Medicine, Houston, TX) by deleting the full coding region of the *Icam1* gene, rendering it null for all isoforms of the receptor. *Icam1*_KO mice were phenotypically normal and did not require special husbandry measures (*Bullard et al., 2007*). Mouse organoids derived from hepatic bipotent stem cells were cultured as previously described (*Blázquez-García et al., 2024*; *Broutier et al., 2016*). Briefly, livers were isolated from mice and digested using collagenase/dispase II (0.125 mg/ml in DMEM/F12 medium). Isolated ducts were mixed with Matrigel (BD Bioscience), seeded, and cultured in 3D. One week after seeding, organoids were mechanically dissociated into small fragments and transferred to fresh Matrigel. Passage was performed weekly at 1:3-1:6 split ratios. For human organoid culture, frozen organoids were previously obtained from healthy liver resections (~1 cm$^3$) from liver transplantations performed at the Erasmus Medical Center, Rotterdam, MEC-2014-060. These organoids were cultured and expanded at the CBM Severo Ochoa under a Material Transfer Agreement between the University of Cambridge and the CBM Severo Ochoa (CSIC-UAM). All in vivo experiments were conducted in compliance with UK legislation set out in the Animal Scientific Procedures Act 1986. All procedures were conducted in accordance with UK Home Office regulations.

## Liver organoid differentiation

The liver organoid differentiation procedure was performed as described elsewhere (*Huch et al., 2015*). Briefly, liver organoids were seeded for 3 d in liver expansion medium (*Broutier et al., 2016*). That medium was then exchanged for differentiation medium (*Broutier et al., 2016*), which was replenished every 2–3 d for a period of 11–13 d. Culture medium was collected 48 hr after the final medium change for albumin secretion assays. Albumin concentration in culture supernatant was determined using the BCG Albumin assay kit (Sigma-Aldrich). For the CFDA secretion assay, organoids were incubated with 0.1 µM of CFDA diluted in phenol red-free DMEM supplemented with 10 mM HEPES, pH 7.4. At 10 min, CFDA fluorescence at the BC-like, lateral lumens of live organoids were detected by time-lapse confocal microscopy under a Nikon AR1 confocal microscope. Time-lapse acquisitions were processed using Fiji image processing software (NIH, Bethesda, MD).

## Method details

### RNA preparation and PCR analysis

RNA was extracted from organoid cultures using an RNeasy Mini RNA Extraction Kit (QIAGEN), and reverse-transcribed using Moloney murine leukemia virus reverse transcriptase (Promega). cDNA was

amplified in a thermal cycler (GeneAmp PCR System 9700, Applied Biosystems). The primers used are listed in Appendix 1—key resources table.

## Cell transfection and stable expression of exogenous proteins

5 µg DNA/$10^6$ cells or 100 nM siRNA /$10^6$ cells were transfected by electroporation (200 mV, 950 µF, and 480 Ω; Bio-Rad). Expression was measured 24–48 hr post-transfection, and the siRNA effect was analyzed 72 hr post-transfection. For stable expression of exogenous proteins, transfected cells were selected by treatment with 0.75 µg/ml G-418 sulfate for at least 4 wk after transfection. Positive cell clones were selected and maintained in a drug-free medium. After several passages in this medium, >80% of cells retained expression of the exogenous protein. For CRISPR/Cas9 gene editing, the cDNA sequence was analyzed using the Breaking-Cas tool (http://bioinfogp.cnb.csic.es/tools/breakingcas), and the selected target sequences were inserted in the pSpCas9(BB)–2A-GFP plasmid, which was a gift from Feng Zhang (Massachusetts Institute of Technology, Cambridge, MA; Addgene plasmid # 48138; http://n2t.net/addgene:48138; RRID:Addgene_48138). GFP-positive cells were sorted after 24 hr of transfection and plated. Individual clones were tested by immunofluorescence and immuno-blot analyses.

## Laser scanning and spinning disk confocal microscopy

HepG2 cells were grown on coverslips (5 × $10^5$ cells/well), fixed in 4% paraformaldehyde (PFA) for 15 min, and rinsed and treated with 10 mM glycine for 2 min to quench the aldehyde groups. Cells were then permeabilized with 0.2% Triton X-100, and rinsed and blocked with 3% bovine serum albumin (BSA) in PBS for 15 min at room temperature (RT). Cells were incubated for 30 min with the primary antibodies, rinsed in PBS, and incubated for 30 min with the appropriate fluorescent secondary antibodies. Actin filaments were detected with fluorophore-conjugated phalloidin (*Ruiz-Sáenz et al., 2011*; see Appendix 1—key resources table). Incubation with antibodies and other fluorescence reagents was always carried out at 37°C. Phosphorylated proteins were stained using Tris-buffered saline (20 mM Tris, pH 7.4; 150 mM NaCl) instead of PBS. For radixin and MRP-2 stainings, cells were fixed with methanol at –20°C for 5 min and then blocked with 10 mM Gly buffer for 15 min at RT before permeabilizing and incubating with specific antibodies as described above. Confocal laser scanning microscopy was carried out with a confocal Zeiss LSM710 system coupled to an AxioImager M2 microscope, a confocal Zeiss LSM 800 system coupled to an AxioObserver microscope, and a confocal Nikon AR1+ system coupled to an Eclipse Ti-E microscope. Confocal spinning microscopy was performed with a Confocal Spinning Disk SpinSR10 system from Olympus coupled to an inVerted microscope IX83. Morphological analyses and BC quantification were performed by staining for F-actin, EBP50, ICAM-1, phospho-MLC, or ZO-1, depending on the experiment. The basolateral and apical intensity of ICAM-1 in WT and ICAM-1_KO HepG2 cells was calculated from confocal images of polarized cell colonies by measuring the fluorescence intensity in Fiji of basolateral and apical areas, respectively. Images were processed with Fiji and Imaris software to enable 3D volume reconstruction. When specified, XZ projections, produced by summing the confocal images containing the structure of interest, were shown.

## STED super-resolution microscopy

STED super-resolution images were acquired using a confocal microscope Leica TCS SP8 (Leica Microsystems) with an immersion objective HCX PL APO CS ×100 NA 1.4 (Leica). For the green channel, sample excitation was achieved using a WLL2 laser settled at 488 nm with 10% of power, fluorescence was collected through a hybrid detector at 498–542 nm, and STED laser beam intensity (592 nm) was used at 30% of its power. For the red channel, samples excitation was achieved using a WLL2 laser settled at 553 nm with 10% of power, fluorescence was collected through a hybrid detector at 563–620 nm, and STED laser beam intensity (660 nm) was used at 60% of its power. Images were acquired at 1024 × 1024 pixels, with a 6× zoom factor (to obtain a pixel size corresponding to 18.36 nm), frame average 1 or 2, in the case of the red channel, a frame accumulation of 4, and a scanning speed of 400 Hz. Under these conditions, no significant photo bleaching was detected.

STED images were analyzed in MATLAB (The MathWorks, Inc, Natick, MA) that detects fluorescent spots and provides their intensity and centroid position through an automatic detection algorithm based on fitting to a Gaussian point-spread function (PSF) profile (*Martínez-Muñoz et al., 2018*). In our

experiments, the PSF profile and its full-width-at-half-maximum ( = 60 nm) describing STED resolution were estimated using fluorescent spots present in the cell cytoplasm, considered as a non-aggregate state of the receptor. This PSF was applied for the identification and analysis of the spot clusters within the biliary channels. The nnd was calculated as the minimum Euclidean distance between each spot and its neighbors. Plot of the mean fluorescence intensities values and statistical analysis, calculated through the unpaired $t$-test with Welch's correction, as obtained using the GraphPad Prism 9 software.

## T-lymphocyte adhesion assays
To measure the ability of T cells to induce their depolarization, HepG2 cells were plated onto 24-well plates ($5 \times 10^4$ cells/well) for 48 hr. The cells were co-cultured with T-lymphocytes at a 2:1 ratio for the indicated time. After washing, cells were fixed, immunofluorescence was performed, and the percentage of CD3-stained T-lymphocytes or CD3-stained T cells adhering to HepG2 cells were measured under a fluorescence microscope.

## Surface protein labeling
HepG2 cells were incubated for 30 min with 250 µg/ml of sulfo-NHS-biotin at 4°C. Cells were then washed and traces of unbound biotin were blocked by incubation for 10 min with DMEM containing 10% FBS. Cells were then incubated with TRITC-conjugated streptavidin at 4°C for 30 min to label surface-biotinylated proteins. Cells were then fixed and immunofluorescence was performed.

## Organoid whole-mount immunostaining
Organoid staining was performed as previously described (*Broutier et al., 2016*). Briefly, for whole-mount immunofluorescent staining, organoids were washed twice in cold PBS before fixing in 4% PFA for 30 min at 4°C. Then, organoids were rinsed in PBS and incubated with gentle agitation in blocking buffer containing 1% DMSO, 2% BSA, and 0.3% Triton X-100. Organoids were rinsed again in PBS and labeled with the primary antibodies at 4°C overnight, rinsed further in PBS, and incubated with the appropriate fluorescent secondary antibodies for 2 h. Finally, nuclei were stained with DAPI.

## Tissue immunofluorescence and immunohistochemistry
Murine livers were removed and fixed overnight in 10% neutral-buffered formalin (Sigma-Aldrich) at RT. After fixation, tissues were incubated in 30% sucrose overnight and then frozen in Tissue-Tek O.C.T. compound. The sections were allowed to cool at RT and then incubated in blocking buffer (1% DMSO, 2% BSA, and 0.3% Triton X-100 in PBS) for 2 hr at RT. Primary antibodies were diluted in blocking buffer and incubated overnight at 4°C. Secondary antibodies were diluted in PBS containing 0.05% BSA and incubated for 2 hr at RT. To facilitate analyses of hepatic ICAM-1 distribution, mice were previously exposed to intraperitoneal 50 µg/kg LPS for 17 hr. The analyses of bile canalicular width were performed in the absence of any inflammatory challenge to reduce the potential interactions of the hepatic receptor with immune cells. More than 80 bile canaliculi from four WT and four *Icam1*_KO mice were quantified.

The immunohistochemical analysis of human hepatic tissue was approved by the Hospital Ethics Committee of the Hospital Universitario de Salamanca. Biopsies from liver allograft rejections and control livers from healthy donors were analyzed. Formalin-fixed, paraffin-embedded sections of 4 µm thickness were deparaffinized in xylene and rehydrated through a decreasing graded ethanol solution series. After suppression of endogenous peroxidase activity (3% hydrogen peroxide, 10 min) and antigen retrieval (boiling in 10 mM citrate buffer, pH 6.0), sections were immunostained with the appropriate primary antibody using the DakoCytomation Envision Plus peroxidase mouse system (Dako). The stained protein was visualized using DAB solution (Dako) and lightly counterstained with Mayer's hematoxylin. To ascertain the specificity of the antibody immunoreactivity, a negative control was carried out in the absence of the primary antibody; this did not produce any detectable immunolabeling.

## Correlative cryo-SXT
HepG2 cells stably expressing GFP-Rab11 were grown on gold finder Quantifoil R2/2 holey carbon grids. To improve the adhesion of cells, the carbon-coated grids were glow-discharged and coated

with FBS before cell seeding. The grids were then vitrified by immersion in liquid ethane on a Leica EM CPC and cryopreserved in liquid nitrogen. Samples selected by cryo-epifluorescence micros-copy were transferred to a Mistral Transmission soft X-ray Microscope (TXM, Alba Synchrotron Light Source). Samples were again mapped using an epifluorescence microscope online with the Mistral TXM. The epifluorescence microscope was equipped with an LED light source (Mightex), a Mitutoyo 20 ×0.42 NA objective (Edmund Optics), and a Retiga CCD detector (QImaging). Coordinates deter-mined in this way were used to select the X-ray acquisition areas. An X-ray energy of 520 eV was used to illuminate the samples. The photon flux on the sample was on the order of $4.6 \times 10^{10}$ photons/s. Datasets were acquired using a Ni zone plate objective lens (ZP) with an outermost zone width of 40 nm (Zeiss) following a tomographic acquisition scheme. The tilt angle of sampling ranged from –65° to 65° at 1° intervals. Exposure time varied with sample thickness from 1 to 5 s per image. The final pixel size was 11.8 nm. Tilt series were normalized relative to the flat-field images, beam current, and exposure time using software developed in-house. Data were also corrected by the apparent point spread function of the microscope (*Otón et al., 2017*) and deconvolved using a Wiener filter with a k = 1/SNR value of 0.1 based on the signal-to-noise ratio (*Otón et al., 2016*). The preprocessed tilt series were aligned to a single rotation axis using IMOD software (*Kremer et al., 1996*) and recon-structed using TOMO3D software with 30 iterations of the SIRT algorithm (*Agulleiro and Fernandez, 2011*). 3D rendering volumes were represented with Chimera and ImageJ (*Schneider et al., 2012*).

## Transmission electron microscopy

Matrigel-embedded liver organoids were in situ fixed with 4% PFA and 2% glutaraldehyde in 0.1 M phosphate buffer, pH 7.4 for 2 hr at RT. Postfixation was carried out with a mixture of 1% osmium tetroxide and 0.08% potassium ferricyanide for 1 hr at 4°C and then with 2% uranyl acetate for 1 hr at RT. Samples were dehydrated with ethanol and processed for standard Epon (TAAB-812) embedding. After polymerization, orthogonal ultrathin sections (80 nm) were collected on formvar-coated slot grids and stained with uranyl acetate and lead citrate. Finally, sections were examined in a Jeol JEM-1400Flash transmission electron microscope operating at 100 kV. Images were recorded with a Gatan OneView (4K × 4K) CMOS camera.

## Biotinylation of ICAM-1-proximal proteins: BioID assay

To generate an expression plasmid containing the ICAM-1-BirA* construct, the sequence coding for GFP in the ICAM-1-GFP was substituted by the BirA* sequence (*Roux et al., 2012*) obtained from the Cav1-BirA* plasmid, kindly provided by Prof. I. Correas (Centro de Biología Molecular Severo Ochoa), with BsrGI and AgeI enzymes (New England Biolabs). The expression vector coding for ICAM-1-BirA* was transfected into HepG2 cells by electroporation and cell clones stably expressing ICAM-1-BirA* were selected with G-418 as previously described (*Reglero-Real et al., 2014*). ICAM-1-BirA* HepG2 cells were cultured on 10-cm-diameter plates and, after 48 hr, incubated with 50 µM biotin for 16 hr, lysed and subjected to a pull-down assay of biotinylated proteins with neutravidin-agarose (Thermo Scientific) as previously described (*Roux et al., 2012*; *Marcos-Ramiro et al., 2016*). In parallel, non-transfected HepG2 cells were used as a control of the precipitation of biotinylated proteins. Lysates and pull-down pellets were analyzed by western blot and LC-ESI QTOF tandem mass spectrometry. The identified proteins were listed and the proteins of interest were validated by western blot.

## Immunoprecipitation assays

HepG2 were transfected with expression vectors coding for FLAG-tagged full-length EBP50 and the indicated FLAG-tagged N-terminal fragments. 48 hr post-transfection, cells were lysed in 400 µl of TNE buffer (50 mM Tris pH 7.4, 150 mM NaCl, 5 mM EDTA) containing 1% Triton-X100 and a protease and phosphatase inhibitor cocktail. Lysates were incubated with 20 µl protein G-coated Sepharose (Sigma-Aldrich) previously conjugated with 5 µl of anti-FLAG antibody by overnight incubation. Lysates were precleared for 1 hr with anti-IG mouse protein-G-sepharose and then incubated with anti-FLAG-conjugated protein-G-sepharose for 3 hr. Antibody-conjugated beads were rinsed in TNE+ TX100 buffer five times and dried by aspiration. Immunoprecipitated proteins were eluted in 20 µl of Laemmli buffer and analyzed by western blot.

## Protein extraction and western blot

Cell lysates and organoid cultures were prepared using Laemmli buffer supplemented with a cocktail of protease inhibitors. The lysates were heated at 95°C for 5 min and cleared by centrifugation at 14 × 10³ rpm for 5 min. The samples were loaded on acrylamide gels and transferred onto an Immobilon-PVDF membrane (Millipore), which was blocked in PBS containing 5% nonfat dry milk and incubated overnight with the indicated primary antibody. Anti-rabbit or anti-mouse horseradish-peroxidase-(HRP)-conjugated secondary antibodies were then used and the antibody–protein complexes were visualized using ECL (GE Healthcare). Band intensities were quantified using Fiji software.

To analyze the solubility of EBP50 and other proteins in non-ionic detergents upon ICAM-1 engagement, cells were incubated with 2 μg/ml anti-ICAM-1 mAb at 40°C for 30 min, washed and incubated at 37°C for the indicated times. Next, cells were lysed at 4°C for 20 min in 400 μl of TNE buffer (50 mM Tris pH 7.4, 150 mM NaCl, 5 mM EDTA) containing 1% Triton-X100 and a protease and phosphatase inhibitor cocktail, passed five times through a 22 G needle, incubated for 5 min at 37°C, and then centrifuged at 14 × 10³ rpm for 5 min at 4°C to separate the supernatant (soluble fraction) from the pellet. Pellets were washed with TNE+ 1% Triton lysis buffer before adding Laemmli buffer to sonicate the samples.

## Quantification and statistical analysis

Data are summarized as the mean and standard deviation (SD) or the mean and standard error of the mean (SEM). Student's two-tailed unpaired samples $t$-tests or two-way ANOVAs were used to establish the statistical significance ($p < 0.05$) of group differences. ANOVAs were applied in *Figures 1F, 4E and 5A–C*. In all cases, data from at least three independent experiments were used. All calculations were performed using Prism 7 software.

## Acknowledgements

We gratefully acknowledge the expert technical advice of the Confocal Microscopy, Electron Microscopy, and Genomic facilities of the CBM Severo Ochoa, especially the contribution of Milagros Guerra, from the electron microscopy facility. We thank the staff of the Advanced Light Microscopy and cryo-EM facilities of the CNB-CSIC for their expert technical assistance with the correlative Cryo-SXT. We also thank Dr. Eva Pereiro at ALBA Synchrotron Light Source (Cerdanyola del Vallès, Spain) for her expert technical advice, Prof. Nancy Hogg at the Francis Crick Institute (London, UK) for generating and sharing the Icam1_KO mouse, Dr. Lucía Cordero Espinoza at the Gurdon Institute (Cambridge, UK) for her technical support in generating liver organoids, and Ana López Sancha for her technical support with isolating the PBMCs. The work was supported by grants PID2020-119881RB-I00 from AEI (to CC-N, CL-P, NC-A, SB, GdR, JF, and JM) and P2022/BMD-7232 TomoXliver2 (to AC, SB, JMC, and JM), and IND2019/BMD-17139 (to JM) from Comunidad de Madrid. This research work was also funded by the European Commission–NextGenerationEU (Regulation EU 2020/2094), through CSIC's Global Health Platform (PTI Salud Global). SB is supported by Endocornea, Convenio Colaboración CSIC, funded by Instituto de Investigación Fundación Jiménez Díaz. CM acknowledges support through the grant PID2021-125386NB-I00 funded by MCIN/AEI/10.13039/501100011033/and FEDER 'ERDF A way of making Europe'. CC-N is a recipient of FPI fellowships from MINECO. NC-A is a recipient of an FPU fellowship from MECD. NR-R is supported by funding from the People Programme (Marie Curie Actions) of the European Union's Seventh Framework Programme (FP7/2007–2013) under REA grant agreement no. 608765 and also by Ramón y Cajal program, grant RYC2021-031221-I and grant PID2022-137552OA-I00 from AEI.

# Additional information

## Funding

| Funder | Grant reference number | Author |
| --- | --- | --- |
| Agencia Estatal de Investigación | PID2020-119881RB-I00 | Cristina Cacho-Navas<br>Carmen López-Pujante<br>Natalia Colás-Algora<br>Susana Barroso<br>Gema de Rivas<br>Jorge Feito<br>Jaime Millán |
| Comunidad de Madrid | P2022/BMD-7232 TomoXliver2 | Ana Cuervo<br>Susana Barroso<br>José-Maria Carazo<br>Jaime Millán |
| Comunidad de Madrid | IND2019/BMD-17139 | Jaime Millán |
| Agencia Estatal de Investigación | PID2021-125386NB-I00 | Carlo Manzo |
| Marie Skłodowska-Curie Actions | 608765 | Natalia Reglero-Real |
| Agencia Estatal de Investigación | PID2022-137552OA-I00 | Natalia Reglero-Real |
| Agencia Estatal de Investigación | RYC2021-031221-I | Natalia Reglero-Real |
| European Commission | Regulation EU 2020/2094 | Jaime Millán<br>José-Maria Carazo<br>Ana Cuervo |
| Fondo Europeo de Desarrollo Regional | PID2021-125386NB-I00 | Carlo Manzo |
| Ministry of Economy and Competitiveness | FPI fellowship | Cristina Cacho-Navas |
| Ministry of Education, Culture and Sports | FPU fellowship | Natalia Colás-Algora |
| Fundación Jiménez Díaz University Hospital Health Research Institute | Endocornea Convenio Colaboración CSIC | Susana Barroso |

The funders had no role in study design, data collection and interpretation, or the decision to submit the work for publication.

## Author contributions

Cristina Cacho-Navas, Data curation, Formal analysis, Investigation, Methodology; Carmen López-Pujante, Natalia Colás-Algora, Susana Barroso, Jorge Feito, Francisca Molina-Jiménez, Pedro Majano, Investigation; Natalia Reglero-Real, Sussan Nourshargh, Meritxell Huch, Resources; Ana Cuervo, Jose Javier Conesa, Alberto Paradela, Germán Andrés, Investigation, Methodology; Gema de Rivas, Investigation, Project administration; Sergio Ciordia, Data curation, Investigation; Gianluca D'Agostino, Data curation, Formal analysis; Carlo Manzo, Data curation, Formal analysis, Visualization; Isabel Correas, Resources, Writing – review and editing; José-Maria Carazo, Resources, Software, Supervision; Jaime Millán, Conceptualization, Supervision, Funding acquisition, Validation, Methodology, Writing - original draft, Writing – review and editing

## Author ORCIDs

Alberto Paradela ⓘ http://orcid.org/0000-0001-6837-7056
Carlo Manzo ⓘ http://orcid.org/0000-0002-8625-0996
Isabel Correas ⓘ https://orcid.org/0000-0002-2286-186X
Jaime Millán ⓘ http://orcid.org/0000-0003-3107-147X

## Ethics

Livers from sacrificed animals were fixed for immunohistochemical analyses and/or their bipotent precursor cells were isolated to generate liver organoids, thus preventing animal experimentation. Wildtype and ICAM-1_KO donor mice on a C57BL/6 background were used in accordance with the institutional Animal Welfare Ethical Review Body (AWERB) and UK Home Office guidelines. ICAM-1_KO mice were kindly provided by Prof. Nancy Hogg (London, UK).

Reviewer #1 (Public Review): https://doi.org/10.7554/eLife.89261.3.sa1

Author response https://doi.org/10.7554/eLife.89261.3.sa2

## Additional files

### Supplementary files

• MDAR checklist

### Data availability

All data generated and analysed during the study are included in the manuscript and supporting files.

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

# Appendix 1

## Appendix 1—key resources table

| Reagent type (species) or resource | Designation | Source or reference | Identifiers | Additional information |
|---|---|---|---|---|
| Gene (*Homo sapiens*) | ICAM1 | GenBank | Gene ID: 3383 | Also known as BB2; CD54; P3.58 |
| Gene (*H. sapiens*) | NHERF1 | GenBank | Gene ID: 9368 | Also known as EBP50; NHERF; NHE-RF; NHERF-1; NPHLOP2; SLC9A3R1 |
| Gene (*Mus musculus*) | Icam1 | GenBank | Gene ID: 15894 | Also known as CD54; Ly-47; Icam-1; MALA-2 |
| Cell line (*H. sapiens*) | HepG2 | ATCC | ATCC HB- 8065 | |
| Cell line (*H. sapiens*) | Human T lymphoblasts | Donors | | Human primary cells prepared from isolated peripheral blood mononuclear cells |
| Transfected construct (*H. sapiens*) | ICAM-1-BirA* | This paper and *Cacho-Navas et al., 2022* | | Backbone pEGFP-N1 |
| Transfected construct (*H. sapiens*) | ICAM-1-GFP | Dr. F. Sánchez- Madrid (Madrid, Spain) (*Barreiro et al., 2002*) | | Backbone pEGFP-N1 |
| Transfected construct (*H. sapiens*) | pSpCas9(BB)-2ª- GFP | PX458 | Addgene; 48138 | Vector for CRISPR-CAS9-medited gene edition Dr. F. Zhang (Cambridge, MA) |
| Transfected construct (*H. sapiens*) | GFP | Clontech | pEGFP-N1 | |
| Transfected construct (*H. sapiens*) | GFP-Rab11 | Dr. F. Martín- Belmonte (Madrid, Spain) (*Rodríguez-Fraticelli et al., 2015*) | | Backbone pEGFP-C1 |
| Transfected construct (*H. sapiens*) | MDR1-GFP | Dr. I.M. Arias, (Bethesda, MD) (*Sai et al., 1999*) | | |
| Transfected construct (*H. sapiens*) | GFP-RhoB | Dr. D. Pérez- Sala (Madrid, Spain) (*Marcos-Ramiro et al., 2016*) | | Backbone pEGFP-C1 |
| Transfected construct (*H. sapiens*) | GFP-Rac1 | Dr. G. Bokoch | Addgene; 12980 | |
| Transfected construct (*H. sapiens*) | GFP-RhoC | From Channing Der (Chapel Hill, NC) | Addgene; 23226 | Backbone pEGFP-C2 |
| Transfected construct (*H. sapiens*) | pCMV-FLAG-EBP50(NHERF1)-FL | Dr. M.M. Georgescu (Houston, TX) | Addgene; 28291 | Backbone pCMV-2-FLAG |
| Transfected construct (*H. sapiens*) | pCMV-FLAG-EBP50(NHERF1)-PDZ1-2 | Dr. M.M. Georgescu (Houston, TX) | Addgene; 28294 | Backbone pCMV-2-FLAG |
| Transfected construct (*H. sapiens*) | pCMV-FLAG-EBP50(NHERF1)-PDZ1iP | Dr. M.M. Georgescu (Houston, TX) | Addgene; 28295 | Backbone pCMV-2-FLAG |
| Transfected construct (*H. sapiens*) | pCMV-FLAG-EBP50(NHERF1)-PDZ2 | Dr. M.M. Georgescu (Houston, TX) | Addgene; 28296 | Backbone pCMV-2-FLAG |
| Transfected construct (*H. sapiens*) | pCMV-FLAG-EBP50(NHERF1)-EB | Dr. M.M. Georgescu (Houston, TX) | Addgene; 28297 | Backbone pCMV-2-FLAG |
| Antibody | Anti-ICAM-1 (mouse monoclonal) | R&D Systems | #BBA3; RRID:AB_356950 | IF 1/400; IP 1/100 |
| Antibody | Anti-ICAM-1 (rabbit polyclonal) | Santa Cruz Biotechnology | sc-7891; RRID:AB_647486 | WB 1/1000 |
| Antibody | Anti-ICAM-1 (mouse monoclonal) | Santa Cruz Biotechnology | sc-107; RRID:AB_627120 | IHC 1/1000 |
| Antibody | Anti-ICAM-1 (rat monoclonal) | EBioscience | 14-0542-81; RRID:AB_529544 | WB 1/1000; IF 1/200 |
| Antibody | Anti-ERK1/2 (rabbit polyclonal) | Santa Cruz Biotechnology | sc-94; RRID:AB_2140110 | WB 1/1000 |
| Antibody | Anti-EBP50 (mouse monoclonal) | Santa Cruz Biotechnology | sc-271552; RRID:AB_10649999 | WB 1/1000; IF 1/400 |
| Antibody | Anti-EBP50 (rabbit polyclonal) | Thermo Fisher Scientific | PA1-090; RRID:AB2191493 | WB 1/1000; IF 1/250 |

*Appendix 1 Continued on next page*

*Appendix 1 Continued*

| Reagent type (species) or resource | Designation | Source or reference | Identifiers | Additional information |
|---|---|---|---|---|
| Antibody | Anti-tubulin (mouse monoclonal) | Santa Cruz Biotechnology | sc-134241; RRID:AB_2009282 | WB 1/5000 |
| Antibody | Anti-SNAP23 (rabbit polyclonal) | Synaptic Systems | 111 202; RRID:AB_887788 | WB 1/1000; IF 1/400 |
| Antibody | Anti-SNAP23 (mouse monoclonal) | Santa Cruz Biotechnology | sc-374215; RRID:AB_10990315 | WB 1/1000 |
| Antibody | Anti-CD59 (mouse monoclonal) | EXBIO | MEM43 (11-233C100); RRID:AB_10735273 | IF 1/400 |
| Antibody | Anti-ERM (rabbit polyclonal) | Cell Signaling Technology | 3142; RRID:AB_2100313 | WB 1/1000; IF 1/400 |
| Antibody | Anti-GFP (mouse monoclonal) | Roche | 11814460001; RRID:AB_390913 | WB 1/1000 |
| Antibody | Anti-ZO-1 (rabbit polyclonal) | Thermo Fisher Scientific | 40-2200; RRID:AB_2533456 | IF 1/500 |
| Antibody | Anti-Rab11 (mouse monoclonal) | Thermo Fisher Scientific | 71-5300; RRID:AB_2533987 | WB 1/1000 |
| Antibody | Anti-TfR (mouse monoclonal) | Thermo Fisher Scientific | 13-6800; RRID:AB_2533029 | WB 1/1000 |
| Antibody | Anti-MLC (rabbit polyclonal) | Cell Signaling Technology | 3672; RRID:AB_10692513 | WB 1/1000; IF 1/200 |
| Antibody | Anti-p(T18/S19)- MLC (rabbit polyclonal) | Cell Signaling Technology | 3671; RRID:AB_330248 | WB 1/1000; IF 1/200 |
| Antibody | Anti-MHC-IIb (rabbit polyclonal) | BioLegend | 909902; RRID:AB_2749903 | IF 1/200 |
| Antibody | Rabbit anti-MHC- IIa (rabbit polyclonal) | BioLegend | 909802; RRID:AB_2734686 | IF 1/200 |
| Antibody | Anti-Exo70 (mouse monoclonal) | Merck (Millipore) | MABT186 clone 70X13F3 | WB 1/500 |
| Antibody | Anti-IgG (mouse monoclonal) | Merck (Sigma-Aldrich) | I5381; RRID:AB_1163670 | IP 1/100 |
| Antibody | Anti-IgG (rabbit polyclonal) | Merck (Sigma-Aldrich) | I5006; RRID:AB_1163659 | IP 1/100 |
| Antibody | Anti-F4/80 (rat monoclonal) | Abcam | ab6640; RRID:AB_1140040 | IF 1/1000 |
| Antibody | Anti-HA-4 antigen (CEACAM) (mouse monoclonal) | University of Iowa | RRID:AB_ 10659875 | IF 1/100 |
| Antibody | Anti-plasmolipin (rabbit polyclonal) | In-house (*Cacho-Navas et al., 2022*) | | IF 1/250 |
| Antibody | Anti-Radixin (rabbit polyclonal) | Cell Signaling | C4G7; RRID:AB_2238294 | IF 1/250 |
| Antibody | Anti-MRP2 (mouse monoclonal) | Enzo Life Sciences | ALX-801-016; RRID:AB_22 73479 | IF 1/250 |
| Antibody | Anti-CD3 OKT3 (mouse monoclonal) | ATCC | CRL-8001 | Purified from mouse hybridomas producing anti-CD3ε mAb OKT3 (our laboratory); IF 1/250 |
| Antibody | Goat F(ab) anti-mouse IgG 596 | Abcam | Abcam (ab6723); RRID:AB_955573 | IF 1/100 |
| Antibody | Donkey anti-mouse Alexa Fluor 488 | Thermo Fisher Scientific | A-21202; RRID:AB_141607 | IF 1/500 |
| Antibody | Donkey anti-mouse Alexa Fluor 555 | Thermo Fisher Scientific | A-31570; RRID:AB_2536180 | IF 1/500 |
| Antibody | Donkey anti-mouse Alexa Fluor 647 | Thermo Fisher Scientific | A-31571; RRID:AB_162542 | IF 1/500 |
| Antibody | Donkey anti-rabbit Alexa Fluor 488 | Thermo Fisher Scientific | A-21206; RRID:AB_141708 | IF 1/500 |

*Appendix 1 Continued on next page*

*Appendix 1 Continued*

| Reagent type (species) or resource | Designation | Source or reference | Identifiers | Additional information |
|---|---|---|---|---|
| Antibody | Donkey anti-rabbit Alexa Fluor 555 | Thermo Fisher Scientific | A-31572; RRID:AB_162543 | IF 1/500 |
| Antibody | Donkey anti-rabbit Alexa Fluor 647 | Thermo Fisher Scientific | A-31573; RRID:AB_2536183 | IF 1/500 |
| Antibody | Donkey anti-rat Alexa Fluor 488 | Thermo Fisher Scientific | A-21208; RRID:AB_2535794 | IF 1/500 |
| Antibody | Donkey anti-mouse HRP | Jackson Immunoresearch | 715-035-151; RRID:AB_2340771 | WB 1/5000 |
| Antibody | Donkey anti-rabbit HRP | GE Healthcare | NA934; RRID:AB_772206 | WB 1/5000 |
| Sequence-based reagent | sgRNA ICAM-1 fw | This paper | Single-guide RNA | CACCGCGCACTCCTGGTCCTGC TCG |
| Sequence-based reagent | sgRNA ICAM-1 rv | This paper | Single-guide RNA | AAACCGAGCAGGACCAGGAGT GCGC |
| Sequence-based reagent | msICAM-1 Fw | This paper | PCR primer | CTTCAACCCGTGCCAAGC |
| Sequence-based reagent | msICAM-1 Rv | This paper | PCR primer | GAAGGCTTCTCTGGGATGGA |
| Sequence-based reagent | msHRPT Fw | This paper | PCR primer | AAGCTTGCTGGTGAAAAGGA |
| Sequence-based reagent | msHRPT Rv | This paper | PCR primer | TTGCGCTCATCTTAGGCTTT |
| Sequence-based reagent | msALB Fw | This paper | PCR primer | GCGCAGATGACAGGGCGGAA |
| Sequence-based reagent | msALB Rv | This paper | PCR primer | GTGCCGTAGCATGCGGGAGG |
| Sequence-based reagent | siControl | Dharmacon | siRNA | AUGUAUUGGCCUGUAUUAGUU |
| Sequence-based reagent | siICAM-1 3'UTR | Dharmacon | siRNA | GAACAGAGUGGAAGACAUAUU |
| Sequence-based reagent | siSLC9A3R1 05 | Dharmacon | siRNA | (TS) CCAGAAACGCAGCAGCAAA |
| Sequence-based reagent | siSLC9A3R1 06 | Dharmacon | siRNA | (TS) GCGAAAACGUGGAGAAGGA |
| Sequence-based reagent | siSLC9A3R1 07 | Dharmacon | siRNA | GCGAGGAGCUGAAUUCCCA |
| Sequence-based reagent | siSLC9A3R1 08 | Dharmacon | siRNA | GAACAGUCGUGAAGCCCUG |
| Peptide, recombinant protein | Streptavidin-Alexa Fluor 555 | Thermo Fisher Scientific | S21381; RRID:AB_2307336 | IF 1/1000 |
| Peptide, recombinant protein | Streptavidin-HRP | Thermo Fisher Scientific | 815-968-0747 | WB 1/10,000 |
| Peptide, recombinant protein | Phalloidin-Alexa Fluor 647 | Thermo Fisher Scientific | A-22287; RRID:AB_2620155 | IF 1/250 |
| Peptide, recombinant protein | PHA | Thermo Fisher Scientific | 10576015 | |
| Peptide, recombinant protein | IL-2 | Thermo Fisher Scientific | PHC0021 | |
| Peptide, recombinant protein | IL-1β | Peprotech | 200-01B | |
| Peptide, recombinant protein | TNF-α | R&D Systems | 210-TA/CF | |
| Peptide, recombinant protein | IFN-γ | Peprotech/Teb- bio | 300-02 | |
| Peptide, recombinant protein | Fibronectin | Corning | 356008 | |
| Commercial assay or kit | BCG albumin assay kit | Merck (Sigma-Aldrich) | MAK124 | |
| Chemical compound, drug | DAPI | Merck | 268298 | IF 1/1000 |
| Chemical compound, drug | Geneticin | Santa Cruz Biotechnology | 29065B | |
| Chemical compound, drug | Sulfo-NHS-biotin | Thermo Fisher Scientific | 21217 | |

*Appendix 1 Continued on next page*

*Appendix 1 Continued*

| Reagent type (species) or resource | Designation | Source or reference | Identifiers | Additional information |
|---|---|---|---|---|
| Chemical compound, drug | Phorbol 12- miristate 13- acetate (PMA) | Merck (Sigma-Aldrich) | P8139 | |
| Chemical compound, drug | Calcein-AM | Thermo Fisher Scientific | C3099 | |
| Chemical compound, drug | Y-27632 | Merck (Calbiochem) | 688000 | |
| Chemical compound, drug | Blebbistatin | Merck (Sigma-Aldrich) | B-0560 | |
| Chemical compound, drug | Biotin | Merck (Sigma-Aldrich) | B4501 | |
| Chemical compound, drug | (5-and-6)-Carboxyfluorescein Diacetate (CFDA) | Thermo Fisher Scientific | C195 | |
| Other | Tissue-Tek O.C.T. | Sakura | 4583 | For preparing liver tissue samples for immunohistochemistry; see 'Tissue immunofluorescence and immunohistochemistry' |
| Other | Holey Carbon Grids for Cryo EM | Quantifoil | R2/2G200F1 | For culturing cells for cryo-soft-X-ray tomography; see 'Correlative cryo-soft X-ray tomography' |
| Other | Neutravidin Agarose | Thermo Fisher Scientific | 29201 | Neutravidin conjugated to solid substrate to perform pull-down assays of biotinylated proteins; see 'Biotinylation of ICAM-1-proximal proteins: BioID assay' |
| Other | Protein-G- sepharose | Rockland | PG50-00-0002 | Protein G-coated Sepharose to conjugate mouse monoclonal antibodies and perform immunoprecipitation; see 'Immunoprecipitation assays' |
| Other | Ficoll | STEMCELL Technologies | 07801 | Reagent for isolating human peripheral blood mononuclear cells (PBMCs) as in *Millán et al., 2002*; see 'Cells and culture' |

