## [Editor Report · eLife assessment]

The authors report **useful** findings on the novel function of apical ICAM-1 in regulating bile duct homeostasis in the liver. The strength of evidence is **solid** using appropriate methodology with only minor weakness. The findings will be of interest to researchers in hepatology and membrane traffic biology.

---

## [Referee Report · Reviewer #1 (Public Review)]

In this study Cacho-Navas et al. describes the role of ICAM-1 expressed on the apical membrane of bile canaliculi and its function to control the homeostasis of the bile canaliculi (BCs). This is a previously unrecognized function of this protein in hepatocytes. The same authors have previously shown that basolateral ICAM-1 plays a role in controlling lymphocyte adhesion to hepatocytes during inflammation and that this interaction is responsible on the loss of polarity of hepatocytes during the disease.

In this new study they show that ICAM-1, is mainly localized in the apical domain of the BC and in association with EBP-50, comunicates with the subapical acto-myosin ring to regulate the size and morphology of the BC.

In this study they used the well-known immortal cell line of liver cells (HepG2) in which they knocked-out ICAM-1 using CRISPR-Cas9 editing and hepatic organoid derived from WT and ICAM-1-KO mice. alternating knocking-out as well as rescue experiments they show that in the absence of apical ICAM-1, the BC dimension and shape are altered.

The conclusions of the study are sufficiently supported by the data.

Comments on revision:

The authors have addressed most of the reviewer's comments in the re-submission, however the use of the organoids as a model to study bile canaliculi is still not convincing.

The HA-4 staining and the space wehere CFDA is secreted does not overlap considering the nuclei position in the middle z-stack section. Also, the interdigitations between cells identified by EM do not form an enclosed space as we should expect for a bile canaliculi.

I understand that other studies have used these organoids to show some hepatocytic functions but at the same time none has characterized before the formation of bile canaliculi as suggested in this study. Therefore a characterization showing the expression of specific markers (i.e mrp2, bsep) should be provided to support this claim.

I would suggest the authors to carefully read the helpful review by Marsee et al., Cell Stem Cell 2021 that clearly and carefully address the classification and validation of liver organoids from experts in the field.

---

## [Author Response]

The following is the authors’ response to the original reviews.

**eLife assessment:**
This valuable study describes a new role of epithelial intercellular adhesion molecule 1 (ICAM-1) protein in controlling bile duct size. The effect is mediated via EBP-50 and subapical actomyosin to regulate size of bile canaliculi. These solid findings have theoretical and practical implications in hepatology and human disorders of bile ducts.
**Public Reviews:**
In this study, Cacho-Navas et al. describe the role of ICAM-1 expressed on the apical membrane of bile canaliculi and its function to control the bile canaliculi (BCs) homeostasis. This is a previously unrecognized function of this protein in hepatocytes. The same authors have previously shown that basolateral ICAM-1 plays a role in controlling lymphocyte adhesion to hepatocytes during inflammation and that this interaction is responsible for the loss of polarity of hepatocytes during disease states.This new study shows that ICAM-1 is mainly localized in the apical domain of the BC and in association with EBP-50, communicates with the subapical acto-myosin ring to regulate the size and morphology of the BC. They used the well-known immortal cell line of liver cells (HepG2) in which they deleted ICAM-1 gene by CRISPR-Cas9 editing and hepatic organoids derived from WT and ICAM-1-KO mice. alternating KO as well as rescue experiments. They show that in the absence of apical ICAM-1, the BC become dilated.The data sufficiently support the conclusions of the study.
**Recommendations for the authors:**

We would like to thank the editor and reviewer for recognizing the manuscript's value and the solid nature of the data. We are also thankful to them for acknowledging that the manuscript supports the conclusions. Below, we have addressed their commentaries and questions in a point-by-point rebuttal document:

We have a few suggestions to improve the manuscript:(1) HepG2 cells form canaliculi-like structures but are not the ideal system to study the apical basal polarity. On the other hand, hepatic organoids can assume a hepatocyte-like phenotype, when cultured under specific conditions but are not functionally comparable to hepatocytes organized in a 3D structure with a hollow lumen that does not recapitulate the BC physiological structure. Therefore, primary hepatocyte in collagen sandwich would be the best model to study the polarization of BCs and could be isolated from WT and ICAM-1-KO mice, that are available. Some of the major findings should be confirmed in this system.

We adopted the culture of hepatic organoids as an experimental strategy motivated by the difficulties to culture primary hepatocytes experienced in previous analyses (RegleroReal, Cell Rep, 2014). The generation of organoids or mature hepatocytes from various sources of stem cells is a commonly employed strategy in hepatocyte cell biology (Meyer et al. EMBO Rep, 2023), due to the difficulties in maintaining mature hepatic epithelial cell cultures for longer than a few hours.

The hepatic organoids we have used in the manuscript are being accepted as advanced cellular strategies for a broad range of fields (Belenguer, Nat Commun, 2022; de Crignis, eLife, 2021; Huch, Cell, 2015). Despite they have some morphological differences with real hepatocytes, we conducted a thorough characterization of their organization identifying canalicular-like structures with functional (CFDA) and molecular (HA-4) markers, which we believe adds value to the manuscript. In addition, the organoid technology has allowed us to import the bipotent precursors to get an permanent source of hepatic cells without the need to import and use the ICAM-1_KO mice, in line with the current guides to reduce animal experimentation.

Taking this into account and to further validate data obtained with our cellular systems, we carried out a quantification of the canalicular diameter in livers from WT and ICAM1_KO cells (New Figure 8B), which validates our data on human cell lines and organoids. We acknowledge that the data obtained from hepatic tissues cannot rule out the contribution of immune cell adhesion to changes in the hepatocyte architecture. However, these experiments, together with the aforementioned organoids and human cell lines, strongly suggest a role for hepatic ICAM-1 in regulating canalicular size.

(2) Overexpression of proteins was used in the study. While this approach is an easier means to visualize, without the use of specific antibodies, it is known to alter the distribution of the protein compared to the endogenous one.

Most of our characterization has been done with antibodies or other fluorescent tools against endogenous proteins localized at BCs: CD59, F-actin, EBP50, MHC, MLC…. In addition, we have included MDR1-GFP and GFP-Rab11, the latter to analyze the subapical compartment (SAC) surrounding BCs. As requested by the reviewer, we now include in a new Supplementary Figure 1C the confocal analyses of endogenous canalicular markers, radixin and MRP2, as well as a new Supplementary Figure 1D containing the staining of an endogenous marker of the SAC, plasmolipin/PLLP (Fraticelli et al, Nat Cell Biol, 2015; Cacho-Navas, Cell Mol Life Sci, 2022), which is consistent with the previous analyses performed with GFP-Rab11.

(3) In the absence of ICAM-1, BCs change shape and dimension but still show the presence of microvilli. What happens to the distribution of polarized transporters like Mrp2, or the transport of bile acids (CFDA clearance) in vivo in the KO animal?

Thank you for this comment. We have analyzed this transporter in murine livers and human hepatic cells. MRP2 distribution does not significantly change and is concentrated in BCs also in ICAM-1_KO livers (New Figure 8C). Likewise, ICAM-1 gene edition does not affect MRP2 localization in the polarized human hepatic epithelial cell line in vitro (Supplementary Figure 1C). We cannot rule out changes for this transporter in other murine liver cell types in vivo, such as sinusoidal endothelial cells, which we believe should be further addressed in a different piece of work.

(4) Does the lack of ICAM-1 affect the cell viability, proliferation or cell size?

ICAM-1_KO cells proliferate slightly more slowly than their WT counterparts, with no detected changes in cell size and death. We present these data in Supplementary Figure 1, A and B.

(5) Are the findings recapitulated in the livers of ICAM-1 KO animals?

ICAM-1 KO animals present enlarged BCs, which is consistent with the main findings of the manuscript (Figure 8B).

The text needs to be more concise. Some of the concepts, in particular those already published, should be condensed. There is a large amount of experiments that are difficult to connect logically. Possibly, cartoons summarizing the approach of the figure could help the reader.

The text of Results and Discussion sections has been shortened by almost 100 words, despite the additional panels and experiments are now described and discussed. New cartoons have been added in Figure 5G and Figure 8F, in addition to those previously included in Figure 1 and Supplementary Figure 6, the latter containing a graphical descriptions of the main conclusions.

Also, more detailed information about statistical analysis (what post-test was used?), concentration of cytokines, and description of the mouse model should be included in the methods.

Cytokine concentrations have been included in the legend of Figure 3 and in the Cell and Culture section of Methods. A brief description of the ICAM-1_KO mouse and the corresponding reference for further information is also provided in the Organoid Culture section of Methods. A statistical analysis section describing the post-test used is also included at the end of Methods. The references of anti-plasmolipin, anti-radixin and antiMRP2 antibodies, as well as the new fixation methods used for immunofluorescence are also included in the corresponding Antibody List and in the Confocal Microscopy section of Methods, respectively ..

Figure 3D. Sample names should be added as in the rest of the figures.

The arrangement of sample names in Figure 3D has been revised and is now similar to that of Figure 3A.